# Long-term outcomes of hospitalized patients with SARS-CoV-2/COVID-19 with and without neurological involvement: 3-year follow-up assessment

Anna Eligulashvili[1], Moshe Gordon[1], Jimmy S. Lee[1], Jeylin Lee[1], Shiv Mehrotra-Varma[1], Jai Mehrotra-Varma[1], Kevin Hsu[2], Imanyah Hilliard[1], Kristen Lee[1], Arleen Li[1], Muhammed Amir Essibayi[3], Judy Yee[1], David J. Altschul[3], Emad Eskandar[3], Mark F. Mehler[4], Tim Q. Duong[1]*

1 Department of Radiology, Montefiore Health System and Albert Einstein College of Medicine, Bronx, New York, United States of America, 2 Department of Radiology, New York University Grossman School of Medicine, New York, New York, United States of America, 3 Department of Neurological Surgery, Montefiore Health System and Albert Einstein College of Medicine, Bronx, New York, United States of America, 4 Department of Neurology, Montefiore Health System and Albert Einstein College of Medicine, Bronx, New York, United States of America

* tim.duong@einsteinmed.edu

**Data Availability Statement:** A minimal data set is publicly available in the online study repository (https://github.com/aeligulash/Long-term-

## Abstract

### Background

Acute neurological manifestation is a common complication of acute Coronavirus Disease 2019 (COVID-19) disease. This retrospective cohort study investigated the 3-year outcomes of patients with and without significant neurological manifestations during initial COVID-19 hospitalization.

### Methods and findings

Patients hospitalized for Severe Acute Respiratory Syndrome Coronavirus 2 (SARS-CoV-2) infection between 03/01/2020 and 4/16/2020 in the Montefiore Health System in the Bronx, an epicenter of the early pandemic, were included. Follow-up data was captured up to 01/23/2023 (3 years post-COVID-19). This cohort consisted of 414 patients with COVID-19 with significant neurological manifestations and 1,199 propensity-matched patients (for age and COVID-19 severity score) with COVID-19 without neurological manifestations. Neurological involvement during the acute phase included acute stroke, new or recrudescent seizures, anatomic brain lesions, presence of altered mentation with evidence for impaired cognition or arousal, and neuro-COVID-19 complex (headache, anosmia, ageusia, chemesthesis, vertigo, presyncope, paresthesias, cranial nerve abnormalities, ataxia, dysautonomia, and skeletal muscle injury with normal orientation and arousal signs). There were no significant group differences in female sex composition (44.93% versus 48.21%, $p = 0.249$), ICU and IMV status, white, not Hispanic (6.52% versus 7.84%, $p = 0.380$), and Hispanic (33.57% versus 38.20%, $p = 0.093$), except black non-Hispanic (42.51% versus 36.03%, $p$

outcomes-of-patients-with-COVID-19-with-neurological-involvement).

**Funding:** The author(s) received no specific funding for this work.

**Competing interests:** The authors have declared that no competing interests exist.

**Abbreviations:** ALT, alanine aminotransferase; ARDS, acute respiratory distress syndrome; AST, aspartate transaminase; BNP, brain natriuretic peptide; BUN, blood urea nitrogen; CDM, Common Data Model; CHF, congestive heart failure; CI, confidence interval; CKD, chronic kidney disease; CNS, central nervous system; COPD, chronic obstructive pulmonary disease; COVID-19, Coronavirus Disease 2019; CRP, C-reactive protein; CT, computed tomography; EMR, electronic medical record; HI, hyperintensities; HR, hazard ratio; ICU, intesive care unit; IMV, invasive mechanical ventilation; INR, international normalized ratio; LDH, lactate dehydrogenase; MACE, major adverse cardiovascular events; MAP, mean arterial pressure; MRI, magnetic resonance imaging; MVD, microvascular disease; OHDSI, Observational Health Data Sciences and Informatics; OMOP, Observational Medical Outcomes Partnership; OR, odds ratio; PCR, polymerase chain reaction; SARS-COV-2, Severe Acute Respiratory Syndrome Coronavirus 2; SBP, systolic blood pressure; SD, standard deviation; SNF, skilled nursing facility; WBC, white blood cell; WM, white matter.

= 0.019). Primary outcomes were mortality, stroke, heart attack, major adverse cardiovascular events (MACE), reinfection, and hospital readmission post-discharge. Secondary outcomes were neuroimaging findings (hemorrhage, active and prior stroke, mass effect, microhemorrhages, white matter changes, microvascular disease (MVD), and volume loss). More patients in the neurological cohort were discharged to acute rehabilitation (10.39% versus 3.34%, $p < 0.001$) or skilled nursing facilities (35.75% versus 25.35%, $p < 0.001$) and fewer to home (50.24% versus 66.64%, $p < 0.001$) than matched controls. Incidence of readmission for any reason (65.70% versus 60.72%, $p = 0.036$), stroke (6.28% versus 2.34%, $p < 0.001$), and MACE (20.53% versus 16.51%, $p = 0.032$) was higher in the neurological cohort post-discharge. Per Kaplan–Meier univariate survival curve analysis, such patients in the neurological cohort were more likely to die post-discharge compared to controls (hazard ratio: 2.346, (95% confidence interval (CI) [1.586, 3.470]; $p < 0.001$)). Across both cohorts, the major causes of death post-discharge were heart disease (13.79% neurological, 15.38% control), sepsis (8.63%, 17.58%), influenza and pneumonia (13.79%, 9.89%), COVID-19 (10.34%, 7.69%), and acute respiratory distress syndrome (ARDS) (10.34%, 6.59%). Factors associated with mortality after leaving the hospital involved the neurological cohort (odds ratio (OR): 1.802 (95% CI [1.237, 2.608]; $p = 0.002$)), discharge disposition (OR: 1.508 (95% CI [1.276, 1.775]; $p < 0.001$)), congestive heart failure (OR: 2.281 (95% CI [1.429, 3.593]; $p < 0.001$)), higher COVID-19 severity score (OR: 1.177 (95% CI [1.062, 1.304]; $p = 0.002$)), and older age (OR: 1.027 (95% CI [1.010, 1.044]; $p = 0.002$)). There were no group differences in radiological findings, except that the neurological cohort showed significantly more age-adjusted brain volume loss ($p = 0.045$) than controls. The study's patient cohort was limited to patients infected with COVID-19 during the first wave of the pandemic, when hospitals were overburdened, vaccines were not yet available, and treatments were limited. Patient profiles might differ when interrogating subsequent waves.

## Conclusions

Patients with COVID-19 with neurological manifestations had worse long-term outcomes compared to matched controls. These findings raise awareness and the need for closer monitoring and timely interventions for patients with COVID-19 with neurological manifestations, as their disease course involving initial neurological manifestations is associated with enhanced morbidity and mortality.

## Author summary

### Why was this study done?

- Neurological symptoms are present in both acute and long-term manifestations of Coronavirus Disease 2019 (COVID-19).

- Patients with acute neurological syndromes during COVID-19 hospitalization are known to have higher short-term mortality rates.

- Although acute outcomes of patients with COVID-19 and neurological manifestations are understood, the long-term sequelae of COVID-19 survivors who suffered acute neurological manifestations are unknown.

### What did the researchers do and find?

- We used 2 cohorts, a neurological group and control group (propensity-matched) both of which were hospitalized for COVID-19, to evaluate long-term outcomes after discharge, up to 3 years later.

- A Kaplan–Meier survival analysis curve was built to analyze the different time-to-death in neurological and control cohorts, revealing that patients in the neurological cohort have shorter time-to-death than patients in the control cohort.

- Brain magnetic resonance imaging (MRI) and computed tomography (CT) studies were scored by radiologists and compared between neurological and control groups, although few group differences in structural abnormalities were observed.

### What do these findings mean?

- Patients who suffer from neurological manifestations during COVID-19 hospitalization have worse long-term outcomes than controls.

- Patients who experienced neurological symptoms during acute COVID-19 infection need to be closely monitored at subsequent follow-up.

- This study came from a single health system with limited sample size.

## Introduction

Severe acute neurological events—such as ischemic stroke, seizures, intracranial hemorrhage and thrombosis, and encephalopathy—have been reported in patients with Coronavirus Disease 2019 (COVID-19) [1–9]. The causes of these central nervous system (CNS) manifestations are multifactorial. There is conflicting evidence concerning whether Severe Acute Respiratory Syndrome Coronavirus 2 (SARS-CoV-2) infects neuronal cells, with some studies reporting neuronal invasion [10], while others report no evidence of direct infection [11,12]. Additional studies suggest that a diffuse microvasculopathy may ensue with endothelial compromise, micro-infarctions, subsequent micro-hemorrhages, and microglial conglomerates with innate immune activation [13–15]. Nonetheless, CNS manifestations could also arise from secondary effects, such as from respiratory distress, cardiovascular stress, sepsis, hypercoagulation, and host-mediated immune responses triggered by SARS-CoV-2 infection. Patients with neurological complications have been shown to have worse acute COVID-19 outcomes including a higher incidence of critical care illness and death compared to propensity-matched controls [16]. Although a few studies have reported postinfection mortality in patients with COVID-19 [17,18], the long-term outcomes of COVID-19 survivors with CNS manifestations are unknown. This question is of particular importance because systemic manifestations of SARS-CoV-2 infection likely form a self-reinforcing loop that amplifies the deleterious effects

of associated brain pathology on overall morbidity and mortality through dynamic nervous system-systemic crosstalk to create a "dyshomeostasis syndrome" [7]. In this context, we have suggested that this process is potentially active during subacute and chronic phases of disease postinfection with profound implications for the occurrence of longer term sequelae, including accelerated aging, neurodegeneration, organ fibrosis, and cancer. The ability to demonstrate that early neurological involvement following SARS-CoV-2/COVID-19 infection predisposes to acute and more long-term particularly severe and life-threatening adverse outcomes is important to demonstrate that ongoing cross-disciplinary brain and body pathological processes are occurring and form a template for informing future more focused and mechanistic studies to temporize or avert profound degrees of morbidity and mortality.

The goal of this study was to evaluate the 3-year outcomes of patients with COVID-19 with significant neurological complaints that warranted neuroimaging during COVID-19 when compared with propensity-matched controls without significant neurological complaints. Improved understanding of the long-term outcomes of patients with COVID-19 with CNS manifestations could help to identify at-risk patients and enable timely interventions to address the potentially high burden of care among these COVID-19 survivors. We hypothesized that patients with COVID-19 with significant neurological complaints have worse outcomes up to 3-years follow-up.

## Methods

### Data sources

This study is reported as per the Strengthening the Reporting of Observational Studies in Epidemiology (STROBE) guideline (S1 Checklist). This is a follow-up retrospective study of a previously reported retrospective cohort study of adult patients [16] admitted to the Montefiore Health System due to COVID-19 between March 01 and April 16, 2020 with confirmed SARS-CoV-2 infection by real-time reverse transcriptase polymerase chain reaction (PCR)-positive assay testing. Follow-up data was captured up to January 23, 2023 (3 years follow-up). The study did not have a prospective protocol or analysis plan. Note that this study cohort was a subset of the patients in the previous paper [18].

The original neurological cohort consisted of 636 hospitalized patients with COVID-19 who experienced various neurological signs and symptoms that warranted neuroimaging during COVID-19 hospitalization. Neurological involvement included acute stroke (confirmed by imaging), new or recrudescent seizures, anatomic brain lesions (subdural hematoma, brain tumor, chronic infarction, or nonspecific lesions), presence of altered mentation with evidence for impaired cognition or arousal, and neuro-COVID-19 complex (headache, anosmia, ageusia, chemesthesis, vertigo, presyncope, paresthesias, cranial nerve abnormalities, ataxia, dysautonomia, and skeletal muscle injury with normal orientation and arousal signs). The original control group consisted of 1,743 patients with COVID-19, hospitalized over the same time period of the neurological group, by 3:1 propensity-matching for age and COVID-19 severity score who did not have significant neurological issues during hospitalization [16] (see below for matching score). After excluding patients who died during hospitalization or were missing from our database, the neurological cohort and control cohort had sample sizes of 414 and 1,199 patients, respectively. Note that the samples differed slightly from the original paper because a few additional patients were found to meet the inclusion/exclusion criteria. Patients who had no electronic medical record (EMR) data after discharge were deemed "lost to follow-up" (did not return to our health system) and excluded from post-discharge analyses. Note that there was 1:1 exact match as defined in the original paper. We did not change the

propensity match criteria. After matching, all demographic variables were not significantly different between groups.

## Data abstraction

Health data were extracted automatically from the electronic medical records as described previously [19–24]. De-identified health data were obtained for research after standardization to the Observational Medical Outcomes Partnership (OMOP) Common Data Model (CDM) version 6. OMOP CDM represents healthcare data from diverse sources, which are stored through standard vocabulary concepts [25], allowing for the systematic analysis of disparate observational databases, including data from the EMR, administrative claims, and disease classifications systems (e.g., International Classification of Disease-10 (ICD-10), Systemized Nomenclature of Medicine–Clinical Terms (SNOMED), Logical Observation Identifiers Names and Codes (LOINC)). ATLAS, a web-based tool developed by the Observational Health Data Sciences and Informatics (OHDSI) community that enables navigation of patient-level, observational data in the CDM format, was used to search vocabulary concepts and facilitate cohort building. Data were subsequently exported and queried as SQLite database files using the DB Browser for SQLite (version 3.12.0). To ensure data accuracy, our team performed extensive cross validation of all major variables extracted by manual chart reviews on subsets of patients [19–24].

## Discharge dispositions

Discharge disposition of survivors from COVID-19 hospitalization were categorized as home (with or without home care), hospice, acute rehabilitation, skilled nursing facility (SNF), and others (i.e., custodial care, supportive care, and psychiatric care).

## Data abstraction

Age, sex, race, ethnicity, comorbidities, and laboratory test data were extracted from electronic medical records at follow-up extending to January 23, 2023. The length of follow-up was defined as the time elapsed between patient COVID-19 hospitalization discharge and either date of death or date of most recent patient encounter up to January 23, 2023. Incidence of stroke, heart attack, major adverse cardiac events (MACE, defined as the composite of cardiovascular death, nonfatal stroke, nonfatal myocardial infarction, new-onset nonfatal heart failure, thromboembolism, and nonfatal cardiogenic shock [26,27]), reinfection, and readmission after COVID-19 discharge for any reason were tabulated at 3-year follow-up. For non-survivors anytime between discharge and January 23, 2023, the cause of death was ascertained retrospectively via chart review and categorized using the primary reason for death as reported on the death certificate.

Preexisting comorbidities included chronic obstructive pulmonary disease (COPD) and asthma, diabetes, congestive heart failure (CHF), and chronic kidney disease (CKD). Intesive care unit (ICU) and invasive mechanical ventilation (IMV) status during primary hospitalization were also reported. Vital signs and laboratory data included temperature, systolic blood pressure (SBP), arterial pressure, D-dimer (DDIM), international normalized ratio (INR), blood urea nitrogen (BUN), creatinine (Cr), sodium, glucose, aspartate transaminase (AST), alanine aminotransferase (ALT), white blood cell (WBC) count, lymphocyte count (Lymph), ferritin (FERR), C-reactive protein (CRP), procalcitonin, lactate dehydrogenase (LDH), brain natriuretic peptide (BNP), troponin-T (TNT), and arterial oxygen saturation. Laboratory data for COVID-19 admission and at follow-up (most recent) were obtained. Results are reported as mean ± standard deviation (SD) or N (%).

## COVID-19 disease severity score

In-hospital COVID-19 disease severity scores [16] were the sum of 5 components that range from 0 to 10 points, on admission: (1) age by decile (age greater than 60, 70, and 80 years earned 1, 2, and 3 points, respectively); (2) mean arterial pressure (MAP) indicating hypotension (MAP below 80, 70, and 60 mmHg earned 1, 2, and 3 points, respectively); (3) oxygen saturation below 94% indicating impaired pulmonary function (1 point); (4) BUN greater than 30 mg/dL indicating impaired renal function (1 point); and (5) INR greater than 1.2 and CRP greater than 10 mg/L, indicating coagulopathy and inflammatory response. Scores ranged from 0 to 10 with higher score reflecting worse COVID-19 disease severity. For controls, propensity matching was performed based on this COVID-19 disease severity score. These scores have not yet been independently validated using external datasets.

## Imaging assessment

Head computed tomography (CT) and brain magnetic resonance imaging (MRI) examinations were assessed at 3 different time points: those obtained before their COVID-19 hospitalization (most recent), during COVID-19 hospitalization, or after COVID-19 hospitalization (most recent) if available. Images were assessed by board-certified radiologists (K.H., J.L., each with at least 10 years of experience) and radiology residents (I.H., K.L., A.L.) under the supervision of the board-certified radiologists, blinded to the patient cohort designation. To establish assessment criteria and the scoring system, our board-certified radiologists and residents worked together to reach consensus by evaluating over a dozen images. Two residents scored each image and at least 1 board-certified radiologist reviewed and adjudicated. Major findings on both CT and MRI were documented for their absence or presence of hemorrhage, active stroke, prior stroke, mass effect, and microhemorrhages. In addition, white matter (WM) change, microvascular disease (MVD), and volume loss on CT and MRI were graded as 0 for normal or not present, 1 for mild, 2 for moderate, and 3 for severe, taking into account the age of patient. MRIs were additionally graded for extent of WM lesions or hyperintensities (HI) using the same grading scale. Findings were tabulated for pre-, intra-, and post-COVID-19 hospitalization. In cases where patients had both CT and MRI at a specific time point, the MRI was used.

Finally, changes in imaging findings before and after COVID-19 hospitalization, if available, were assessed using the grading scale of 0: no change, ±1: mild change, ±2: moderate change, and ±3 severe change, with positive changes indicating worsening and negative changes indicating improvement between the 2 time points. Pre-COVID-19 images were used as a baseline if both pre- and intra-COVID-19 images were available.

## Primary outcomes

Primary outcomes were mortality, stroke, heart attack, MACE, reinfection, and hospital readmission post-discharge until January 23, 2023 (3 years post COVID-19). Secondary outcomes, measured after COVID-19 hospitalization discharge, were qualitative and score-based clinical neuroimaging findings, which included the absence or presence of hemorrhage, active stroke, prior stroke, mass effect, and microhemorrhages, as well as scores of WM changes, MVD, and volume loss.

Associative models using univariate logistic regression were employed to identify variables associated with mortality after discharge. Input for age was a continuous variable. Input for discharge disposition status was a single variable. Input for COVID-19 severity scores was a continuous variable. The remainder of the variables used in the model were categorical variables. No variables were adjusted for in either model. Odds ratios (OR) and 95% confidence interval (CI) were computed. In addition, Kaplan–Meier curves were constructed and

analyzed using GraphPad Prism, with the outcome event being classified as dead (death date) or alive (most recent patient encounter). The survival curves of the neurological and control cohorts were compared with the logrank test (Mantel–Cox) method, resulting in a log rank hazard ratio (HR), 95% CI, and $p$-value.

## Statistical analysis

Analysis of group differences of demographic and clinical variables employed $\chi^2$ tests for categorical variables and two-tailed $t$ tests for continuous variables via the statistical library in SPSS and RStudio, respectively. Analysis of associative variables was performed in Rstudio using a univariate logistic regression model. Statistical comparison of imaging scores and changes of scores from baseline (pre- or intra-COVID-19 hospitalization) to follow-up (post-COVID-19 hospitalization) employed unpaired $t$ test. $P < 0.05$ was considered statistically significant unless otherwise specified. Statistics were not adjusted for multiple comparisons due to the exploratory nature of the study.

## Ethics statement

This retrospective study using real-world data was approved by the Montefiore Einstein Institutional Review Board (#2021–13658) with a waiver of informed consent. All methods were performed in accordance with relevant guidelines and regulations pertaining to human subjects.

## Results

Of the original neurological cohort of 636 patients, 414 were discharged alive and 371 returned to our health system. Of the original control cohort of 1,743, 1,199 were discharged alive and 1,071 returned to our health system (**S1 Fig**). The results of the propensity match are available in **S1 Table**. The overall attrition rate was approximately 12.3%. The average length of follow-up was 602 ± 400 and 672 ± 269 days (mean ± SD) for the neurological and control cohorts, respectively. The average range of follow-up was (minimum 1, maximum 1,037 days) and (minimum 1, maximum 1,043 days) for the neurological and control cohorts, respectively.

## Discharge disposition

**Fig 1** shows the discharge dispositions of the neurological and control cohorts stratified by COVID-19 severity score. Patients with high severity scores were less likely to be discharged home and more likely to be discharged to an SNF or hospice in both groups. However, there were relatively more patients discharged to SNF and relatively fewer patients discharged to home in the neurological cohort compared to the control cohort.

## Primary outcomes of neurological vs. control cohort

**Table 1** shows the profiles of the survivors at discharge grouped by neurological and control cohorts. There were no statistically significant group differences in female sex composition (186 (44.93%) versus 578 (48.21%), $p = 0.249$), all major comorbidities, ICU and IMV status, and race and ethnicity, except black non-Hispanic (176 (42.51%) versus 432 (36.03%), $p = 0.019$). More patients in the neurological cohort were discharged to acute rehabilitation (43 (10.39%) versus 40 (3.34%), $p < 0.001$) and SNF (148 (35.75%) versus 304 (25.35%), $p < 0.001$) and fewer survivors in the neurological cohort were discharged to home (208 (50.24%) versus 799 (66.64%), $p < 0.001$) compared to the control cohort. With respect to laboratory data, there were few group differences between those at admission and at follow-up, as well as between groups (**S2 Table**). Patients in the neurological cohort had lower temperature,

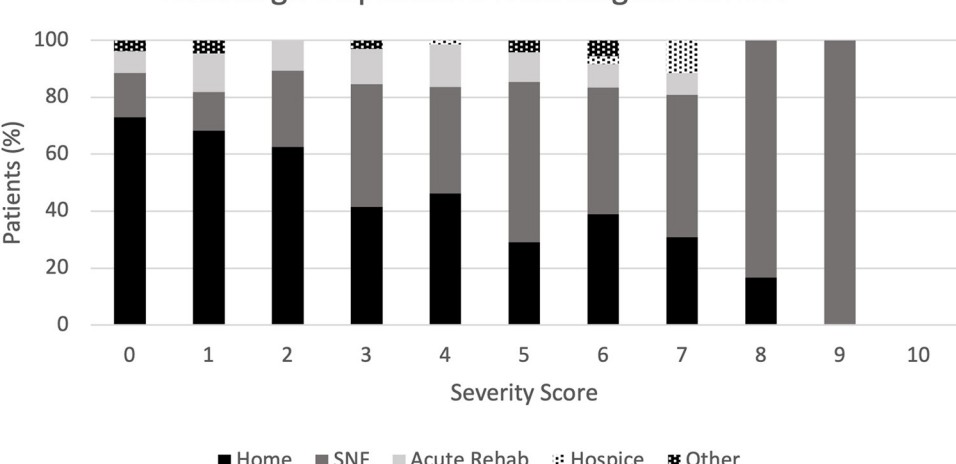

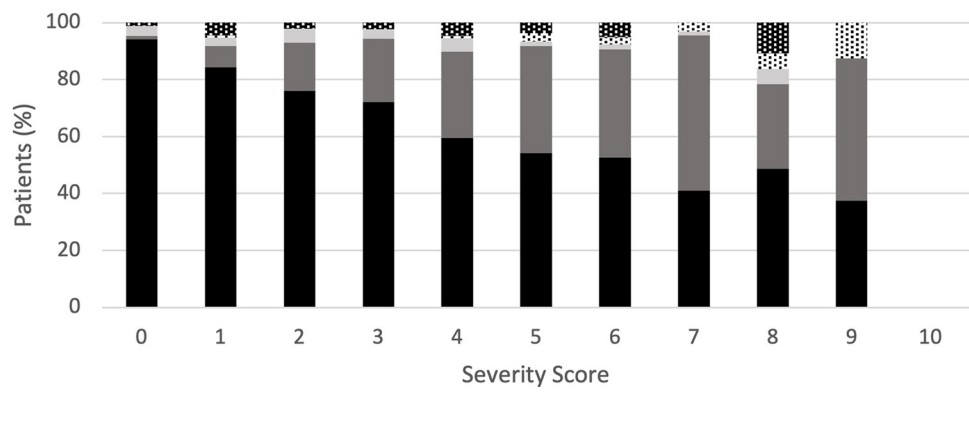

**Fig 1. Percent of patients in the neurological and control cohorts discharged to home, acute rehabilitation, SNF, hospice, and others for different COVID-19 severity score.** "Other" included custodial care, supportive care, and psychiatric care. COVID-19, Coronavirus Disease 2019; SNF, skilled nursing facility.

AST, ALT, CRP, higher mean arterial pressure, DDIM, BUN, glucose, procalcitonin, BNP, TNT, and pulse oximetry at admission, and higher SBP, BUN, Cr, and BNP at follow-up than patients in the control cohort.

Incidence of readmission (272 (65.70%) versus 728 (60.72%), $p = 0.036$), stroke (26 (6.28%) versus 28 (2.34%), $p < 0.001$), and MACE (85 (20.53%) versus 198 (16.51%), $p = 0.032$) were significantly higher in the neurological cohort than the control cohort. There were however no significant group differences in heart attack ($p = 0.166$) and reinfection ($p = 0.420$) after discharge. Mortality rates post-COVID-19 hospitalization were also higher in the neurological cohort compared to the control cohort at 0.5 years (32 (7.73%) versus 44 (3.67%), $p < 0.001$), 1 year (38 (9.18%) versus 59 (4.92%), $p < 0.001$), and 3 years (58 (14.01%) versus 94 (7.84%), $p < 0.001$) follow-up. Kaplan–Meier survival analysis (**Fig 2**) showed that the neurological cohort had a significantly lower time to death than the control cohort at all time points (HR: 2.346 (95% CI [1.586, 3.470]; $p < 0.001$)). The number of patients treated in the ICU or with IMV during their primary

**Table 1. Demographics, comorbidities, and outcomes of survivor patients in the neurological and control cohorts.** Mean ± SD or *n* (%). χ² used to compare categorical variables and two-tailed *t* test use to compare continuous variables between neurological and control groups.

| | Neurological cohort (N = 414) | Control cohort (N = 1,199) | P-value |
|---|---|---|---|
| **Patient characteristics** | | | |
| Age, years old (at admission) | 69.71 ± 15.80 | 70.26 ± 15.14 | 0.539 |
| Female sex (%) | 186 (44.93%) | 578 (48.21%) | 0.249 |
| Combined race and ethnicity | | | |
| White, not Hispanic | 27 (6.52%) | 94 (7.84%) | 0.380 |
| Black, not Hispanic | 176 (42.51%) | 432 (36.03%) | **0.019** |
| Hispanic | 139 (33.57%) | 458 (38.20%) | 0.093 |
| Other | 62 (14.98%) | 169 (14.10%) | 0.659 |
| **Comorbidities** (at admission) | | | |
| Hypertension | 186 (44.93%) | 523 (43.62%) | 0.644 |
| COPD/asthma | 42 (10.14%) | 144 (12.01%) | 0.306 |
| Diabetes | 126 (30.43%) | 343 (28.61%) | 0.480 |
| CHF | 49 (11.84%) | 118 (9.84%) | 0.251 |
| CKD | 90 (21.74%) | 223 (18.60%) | 0.164 |
| **COVID-19 severity** | | | |
| ICU | 30 (7.25%) | 77 (6.42%) | 0.604 |
| IMV | 24 (5.80%) | 59 (4.62%) | 0.522 |
| **Discharge disposition** | | | |
| Home | 208 (50.24%) | 799 (66.64%) | **<0.001** |
| Acute rehab | 43 (10.39%) | 40 (3.34%) | **<0.001** |
| SNF | 148 (35.75%) | 304 (25.35%) | **<0.001** |
| Hospice | 6 (1.45%) | 15 (1.25%) | 0.380 |
| Other | 9 (2.17%) | 41 (3.42%) | 0.104 |
| **Outcomes (January 10, 2023)** | | | |
| Hospital readmission | 272 (65.70%) | 728 (60.72%) | 0.036 |
| Stroke | 26 (6.28%) | 28 (2.34%) | **<0.001** |
| Heart attack | 22 (5.31%) | 50 (4.17%) | 0.166 |
| MACE | 85 (20.53%) | 198 (16.51%) | **0.032** |
| Mortality after discharge | | | |
| 0.5 years | 32 (7.73%) | 44 (3.67%) | **<0.001** |
| 1.0 years | 38 (9.18%) | 59 (4.92%) | **<0.001** |
| 3.0 years | 58 (14.01%) | 94 (7.84%) | **<0.001** |
| SARS-CoV-2 reinfection | 18 (4.35%) | 55 (4.59%) | 0.420 |

CHF, congestive heart failure; CKD, chronic kidney disease; COPD, chronic obstructive pulmonary disease; ICU, intensive care unit; IMV, invasive mechanical ventilation; MACE, major adverse cardiac event; SD, standard deviation; SNF, skilled nursing facility.

hospitalization who subsequently experienced each post-discharge outcome is reported in **S3 Table**. Of 414 patients and 1,043 patients in the neurological and control cohorts, respectively, who were readmitted post-COVID, 30 (7.25%) neurological and 77 (7.38%) control had been in the ICU and 24 (5.80%) neurological and 59 (5.66%) control had IMV.

## Outcomes versus COVID-19 severity scores

**S2A Fig** shows the distribution of COVID-19 severity scores in each cohort, confirming proper propensity matching by severity score among survivors after COVID-19

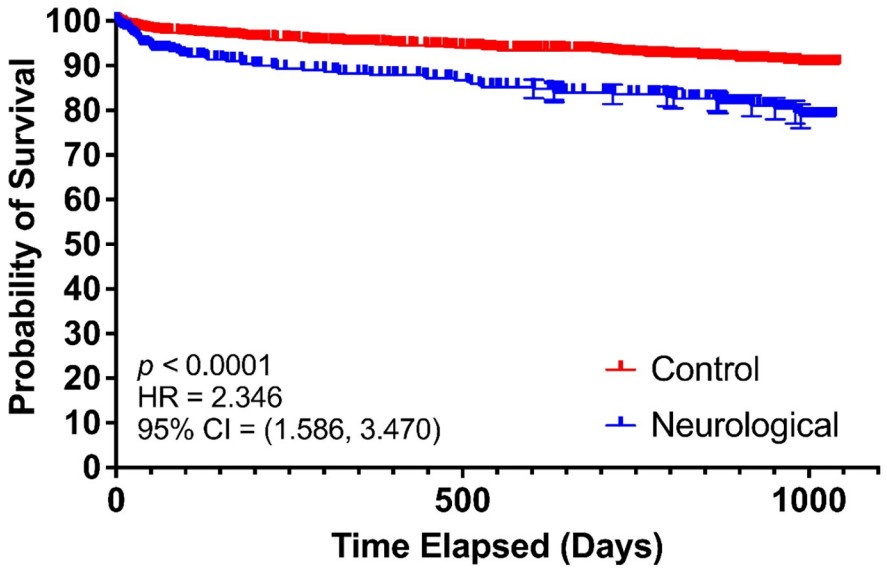

**Fig 2. Kaplan–Meier survival curve analyzing the probability of survival after discharge from COVID-19 hospitalization in the neurological cohort versus the control cohort.** Number at risk at 90-day time intervals provided. HR = 2.346, 95% CI = (1.586, 3.470), *p*-value <0.0001. Whiskers shows standard error. CI, confidence interval; COVID-19, Coronavirus Disease 2019; HR, hazards ratio.

hospitalization discharge. Primary outcomes were analyzed with respect to COVID-19 severity score for survivors and non-survivors post-discharge (**S2B–S2F Fig**). Readmission for any medical reasons were similar among all severity scores for both cohorts. Incidence of stroke was high for all severity scores in the neurological cohort but was generally lower for matching scores in the control cohort. Incidence of heart attack generally appeared to grow as severity score increased for both cohorts. The percent of patients who had MACE was distributed over a range of scores for both cohorts, with higher COVID-19 severity scores revealing a slightly higher percentage of patients with MACE at follow-up. Non-survivors at follow-up had higher COVID-19 severity score compared to survivors.

## Cause of death post COVID-19 discharge

There were no group differences in cause of death between patients belonging to neurological and control cohorts. Beside the unknown cause of death (27.59% neurological versus 34.07% control, *p* = 0.485), the major causes of death after discharge were heart disease (13.79% versus 15.38%, *p* = 0.851), sepsis (8.62% versus 17.58%, *p* = 0.145), influenza and pneumonia (13.79%

**Table 2. Primary cause of death of patients who died after discharge up to January 23, 2023.** *P*-value for difference between the neurological and control cohorts. χ²
used to compare categorical variables between groups.

| Cause of death after discharge | Neurological (*N* = 58) | Control (*N* = 94) | *P*-value | 95% CI of mean/proportion difference |
|---|---|---|---|---|
| Unknown | 16 (27.59%) | 31 (34.07%) | 0.485 | (−20.31%, 9.53%) |
| Heart disease | 8 (13.79%) | 14 (15.38%) | 0.851 | (−12.53%, 10.33%) |
| Sepsis | 5 (8.62%) | 16 (17.58%) | 0.145 | (−18.88%, 2.08%) |
| Influenza and pneumonia | 8 (13.79%) | 9 (9.89%) | 0.423 | (−6.46%, 14.90%) |
| COVID-19 | 6 (10.34%) | 7 (7.69%) | 0.535 | (10.34%, 7.45%) |
| ARDS, Hypoxia | 6 (10.34%) | 6 (6.59%) | 0.379 | (−5.30%, 13.23%) |
| Other (GI bleed, pancreatitis, ICH, acute liver failure) | 3 (5.17%) | 6 (6.59%) | 0.759 | (5.17%, 6.38%) |
| Cancer | 2 (3.45%) | 2 (2.20%) | 0.621 | (3.45%, 2.13%) |
| Nephritis, nephrotic syndrome, and nephrosis | 2 (3.45%) | 1 (1.10%) | 0.305 | (3.45%, 1.06%) |
| Stroke (cerebrovascular diseases) | 1 (1.72%) | 1 (1.10%) | 0.729 | (1.72%, 1.06%) |
| Accidents (unintentional injuries) | 1 (1.72%) | 1 (1.10%) | 0.729 | (1.72%, 1.06%) |

ARDS, acute respiratory distress syndrome; CI, confidence interval; GI, gastrointestinal; ICH, intracranial hemorrhage.

versus 9.89%, *p* = 0.423), COVID-19 (10.34% versus 7.69%, *p* = 0.535), and acute respiratory
distress syndrome (ARDS) (10.34% versus 6.59%, *p* = 0.379) (**Table 2**).

## Survivors and non-survivors post-discharge

**Table 3** compares the profiles of survivors and non-survivors in the neurological and control
cohorts post-discharge. There were few differences of survivors and non-survivors between
cohorts. Both within the neurological and control cohorts, non-survivors were significantly
older (neuro: 76.40 ± 11.71 versus 67.85 ± 15.85, *p* < 0.001 and control: 76.56 ± 11.23 versus
69.20 ± 14.95, *p* < 0.001) compared to survivors. In the control cohort, non-survivors addi-
tionally had higher COVID-19 severity score (control: 4.74 ± 2.05 versus 3.36 ± 2.06,
*p* < 0.001) than survivors. Although no differences in comorbidities were observed in the neu-
rological cohort, control non-survivors had higher incidence of hypertension (57.45% versus
48.05%, *p* = 0.047), diabetes (41.49% versus 31.40%, *p* = 0.044), CHF (32.98% versus 8.75%,
*p* < 0.001), and CKD (38.30% versus 19.49%, *p* < 0.001) compared to control survivors. Both
neurological and control non-survivors were less likely to be discharged home and more likely
to be discharged to SNF. There were no significant differences in ICU and IMV status between
any groups.

## Risk factors for mortality after discharge

A univariate logistic regression model found 5 significant variables associated with mortality
post-discharge (**Table 4**). These variables included belonging to the neurological cohort (OR:
1.802 (95% CI [1.237, 2.608]; *p* = 0.002)), discharge disposition (OR: 1.508 (95% CI [1.276,
1.775]; *p* < 0.001)), congestive heart failure (OR: 2.281 (95% CI [1.429, 3.593]; *p* < 0.001)),
COVID-19 severity score (OR: 1.177 (95% CI [1.062, 1.304]; *p* = 0.002)), and age (OR: 1.027
(95% CI [1.010, 1.044]; *p* = 0.002)). Male sex (OR: 1.387 (95% CI [0.963, 2.004]; *p* = 0.080)) was
statistically insignificant.

## Imaging findings

**Table 5** summarizes the pre-, intra-, and post-COVID-19 neuroradiological findings. The
number of patients who underwent imaging procedures varied between pre-, intra-, and post-

**Table 3. Demographics and comorbidities of patients who died versus survived after discharge in the neurological and control cohorts.** Mean ± SD or N (%). $\chi^2$ used to compare categorical variables and two-tailed $t$ test use to compare continuous variables between groups.

| | Neurological cohort (N = 371) | | | Control cohort (N = 1,043) | | | p-val (survivors) | p-val (non-survivors) |
|---|---|---|---|---|---|---|---|---|
| | Survivors N = 313 (84.37%) | Non-survivors N = 58 (15.63%) | p-val | Survivors N = 949 (90.99%) | Non-survivors N = 94 (9.01%) | p-val | | |
| Age, years old (admission) | 67.85 ± 15.85 | 76.40 ± 11.71 | **<0.001** | 69.20 ± 14.95 | 76.56 ± 11.23 | **<0.001** | 0.253 | 0.707 |
| Female sex | 147 (46.96%) | 22 (37.93%) | 0.212 | 480 (50.58%) | 43 (45.74%) | 0.393 | 0.580 | 0.462 |
| **Combined race and ethnicity** | | | | | | | | |
| White, non-Hispanic | 23 (7.35%) | 3 (5.17%) | 0.350 | 68 (7.17%) | 9 (9.57%) | 0.449 | 0.957 | 0.218 |
| Black, non-Hispanic | 129 (41.21%) | 28 (48.28%) | 0.341 | 350 (36.88%) | 42 (44.68%) | 0.133 | 0.051 | 0.535 |
| Hispanic | 106 (33.87%) | 19 (32.76%) | 0.802 | 383 (40.36%) | 39 (41.49%) | 0.855 | 0.052 | 0.537 |
| Other | 47 (12.02%) | 5 (8.62%) | 0.321 | 139 (14.65%) | 3 (3.19%) | **0.004** | 0.959 | 0.187 |
| **COVID-19 Severity** | | | | | | | | |
| ICU | 26 (8.31%) | 4 (6.90%) | 0.717 | 65 (6.85%) | 7 (7.45%) | 0.827 | 0.387 | 0.899 |
| IMV | 20 (6.39%) | 4 (6.90%) | 0.885 | 50 (5.27%) | 5 (5.32%) | 0.983 | 0.452 | 0.689 |
| Severity score | 3.32 ± 2.02 | 4.40 ± 1.79 | 0.088 | 3.36 ± 2.06 | 4.74 ± 2.05 | **<0.001** | 0.034 | 0.004 |
| **Comorbidities** (at admission) | | | | | | | | |
| Hypertension | 151 (48.24%) | 31 (53.45%) | 0.147 | 456 (48.05%) | 54 (57.45%) | **0.047** | 0.891 | 0.989 |
| COPD/asthma | 35 (11.18%) | 6 (10.34%) | 0.852 | 128 (13.49%) | 17 (18.09%) | 0.123 | 0.539 | 0.196 |
| Diabetes | 100 (31.95%) | 21 (36.21%) | 0.213 | 298 (31.40%) | 39 (41.49%) | **0.044** | 0.997 | 0.823 |
| Congestive heart failure | 38 (12.14%) | 11 (18.97%) | 0.478 | 83 (8.75%) | 31 (32.98%) | **<0.001** | 0.104 | **0.025** |
| Chronic kidney disease | 75 (23.96%) | 14 (24.14%) | 0.676 | 185 (19.49%) | 36 (38.30%) | **<0.001** | 0.076 | 0.088 |
| **Discharge disposition** | | | | | | | | |
| Home | 173 (55.27%) | 17 (29.31%) | **<0.001** | 686 (72.29%) | 38 (40.43%) | **<0.001** | **<0.001** | **0.083** |
| Acute rehab | 33 (10.54%) | 7 (12.07%) | 0.365 | 36 (3.79%) | 4 (4.26%) | 0.390 | **<0.001** | **0.035** |
| SNF | 96 (30.67%) | 32 (55.17%) | **<0.001** | 203 (21.39%) | 39 (41.49%) | **<0.001** | **<0.001** | 0.050 |
| Hospice | 2 (0.64%) | 2 (3.45%) | **0.029** | 1 (0.11%) | 4 (4.26%) | **<0.001** | **<0.001** | 0.402 |
| Other | 9 (2.88%) | 0 (0.00%) | | 23 (2.42%) | 9 (9.57%) | **<0.001** | 0.303 | |

COPD, chronic obstructive pulmonary disease; COVID-19, Coronavirus Disease 2019; ICU, intensive care unit; IMV, invasive mechanical ventilation; p-val, p-value; SD, standard deviation; SNF, skilled nursing facility.

COVID-19. Of those with imaging, about 20% were MRI and 80% were CT. For qualitative assessment, 30% to 40% of all patients had prior strokes for all 3 time points, whereas the presence of hemorrhage, active stroke, and/or mass effect were relatively low (0% to 10% with most around 5%). There were, however, no statistically significant group differences between neurological and control cohorts in these qualitative findings at all 3 time points.

For score-based assessment, the average scores for age-appropriate WM change and MVD were about 1 (mild abnormality), and the average scores for age-appropriate volume loss and WM lesions were about 0.5 (no to mild abnormality). Distribution of scores were similar between neurological and control groups. There were no differences in scores between groups, except for volume loss post-COVID-19 (average score: 0.72 ± 0.71 neurological versus 0.57 ± 0.69 control, $p = 0.045$, and score of 0: 42.36% neurological versus 53.78% control, $p = 0.037$).

**Table 6** shows the changes in imaging findings between baseline (pre- or intra-COVID-19) and follow-up (post-COVID-19). There were significant increases in the incidence of active stroke (neurological: 6.25%, $p = 0.021$; control: 3.82%, $p = 0.039$), prior stroke (12.5%, $p < 0.001$; 9.55%, $p = 0.002$), and microhemorrhages (7.14%, $p = 0.012$; 6.37%, $p = 0.004$)

**Table 4. ORs of mortality post-discharge.** Patients lost to follow-up were excluded.

|  | OR | 95% CI | *p*-val |
|---|---|---|---|
| Belonging to neurological cohort | 1.802 | (1.237, 2.608) | **0.002** |
| Discharge disposition | 1.508 | (1.276, 1.775) | **<0.001** |
| Congestive heart failure | 2.281 | (1.429, 3.593) | **<0.001** |
| COVID-19 severity score | 1.177 | (1.062, 1.304) | **0.002** |
| Age | 1.027 | (1.010, 1.044) | **0.002** |
| Male sex | 1.387 | (0.963, 2.004) | 0.080 |
| Chronic kidney disease | 1.321 | (0.849, 2.039) | 0.213 |
| Diabetes | 1.131 | (0.744, 1.710) | 0.561 |
| COPD/asthma | 1.130 | (0.651, 1.892) | 0.652 |
| Hypertension | 0.985 | (0.649, 1.492) | 0.942 |

CI, confidence interval; COPD, chronic obstructive pulmonary disease; COVID-19, Coronavirus Disease 2019; OR, odds ratio; *p*-val, *p*-value.

between the 2 time points, indicative of age-related effects. There were, however, no group differences ($p > 0.05$ for all).

WM changes (neurological: $0.12 \pm 0.44$, $p = 0.006$; control: $0.18 \pm 0.46$, $p < 0.001$), MVD ($0.15 \pm 0.41$, $p < 0.001$; $0.18 \pm 0.46$, $p < 0.001$) and volume loss ($0.15 \pm 0.41$, $p < 0.001$; $0.17 \pm 0.42$, $p < 0.001$) were higher (worsening) post-COVID compared to baseline for both neurological and control cohorts. There were, however, also no group differences ($p > 0.05$ for all).

## Discussion

This study investigated the 3-year outcomes of hospitalized patients with COVID-19 with and without major neurological issues at initial hospital presentation. The major findings are: (1) patients with COVID-19 with significant neurological issues that warranted neuroimaging were more likely to be discharged to acute rehabilitation and skilled nursing facilities compared to matched controls; (2) the neurological cohort had higher mortality rates after discharge compared to controls; (3) the incidence of readmission, stroke, and MACE, but not heart attack or reinfection, were higher in the neurological cohort at 3 years follow-up; (4) the primary causes of death after discharge for both cohorts was unknown, followed by heart failure, sepsis, influenza and pneumonia, COVID-19, and ARDS; (5) patients who died post-discharge were significantly older, had higher COVID-19 severity score (in the control cohort), and were more likely to have been discharged to skilled nursing facilities at discharge compared to survivors; (6) there were no group differences in general radiological findings with respect to hemorrhage and stroke, although the neurological cohort showed significantly more age-appropriate volume loss than the control cohort.

Approximately half of the patients in the neurological cohort and one-thirds of the patients in the control cohort were discharged to SNF, acute rehabilitation, or hospice. These findings indicated that many patients with COVID-19 were not functionally independent after discharge [28–32], especially those in the neurological cohort. Few studies today have reported home, SNF, and acute rehabilitation discharge rates after COVID-19 hospitalization [28–32]. These findings suggest that patients in the neurological cohort likely needed more follow-up medical care at discharge.

About 65% and 60% of all patients in the neurological and control cohorts, respectively, were readmitted to the health system for medical reasons over 3 years. This is not surprising

**Table 5. Imaging findings pre, during, and post-COVID-19 hospitalization.** The neurological cohort had pre- and post-COVID-19 imaging at 456 ± 729 days and 335 ± 274 days before and after hospitalization, respectively. The control cohort had pre- and post-COVID-19 imaging at 828 ± 974 and 432 ± 300 days before and after hospitalization, respectively. Patients who died during COVID-19 hospitalization are excluded. Imaging studies were scored as 0 (normal or no abnormality), 1 (mild abnormality), 2 (moderate abnormality), and 3 (severe abnormality). Mean ± SD or $n$ (%). $\chi^2$ used to compare categorical variables and two-tailed $t$ test use to compare continuous variables between groups.

| | | Pre COVID-19 | | | During COVID-19 | | | Post COVID-19 | | |
|---|---|---|---|---|---|---|---|---|---|---|
| | | Neurological | Control | *p*-val | Neurological | Control | *p*-val | Neurological | Control | *p*-val |
| Total patients | | 94 | 143 | | 65 | 32 | | 144 | 251 | |
| MRI | | 21 (22.34%) | 32 (22.38%) | 1.000 | 6 (9.23%) | 6 (18.75%) | 0.312 | 32 (21.92%) | 67 (26.69%) | 0.386 |
| CT | | 73 (77.66%) | 111 (77.62%) | 1.000 | 59 (90.77%) | 26 (81.25%) | 0.312 | 112 (76.71%) | 184 (73.31%) | 0.386 |
| **Qualitative findings** | | | | | | | | | | |
| Hemorrhage | | 4 (4.26%) | 3 (2.10%) | 0.570 | 5 (7.69%) | 0 (0%) | 0.262 | 7 (4.86%) | 7 (2.79%) | 0.430 |
| Active stroke | | 6 (6.38%) | 7 (4.90%) | 0.841 | 7 (10.77%) | 2 (6.25%) | 0.727 | 11 (7.64%) | 12 (4.78%) | 0.345 |
| Prior stroke | | 35 (37.23%) | 43 (30.07%) | 0.314 | 23 (35.38%) | 11 (34.38%) | 1.000 | 58 (40.28%) | 76 (30.28%) | 0.056 |
| Mass effect | | 2 (2.13%) | 7 (4.90%) | 0.457 | 6 (9.23%) | 2 (6.25%) | 0.913 | 10 (6.94%) | 14 (5.58%) | 0.743 |
| Microhemorrhage | | 5 (5.32%) | 8 (5.59%) | 1.000 | 5 (4.62%) | 0 (0%) | 0.262 | 9 (6.25%) | 17 (6.77%) | 1.000 |
| **Scoring findings** | | | | | | | | | | |
| White matter change | | 0.99 ± 0.81 | 0.93 ± 0.75 | 0.572 | 0.97 ± 0.78 | 1.13 ± 0.89 | 0.411 | 1.06 ± 0.86 | 1.05 ± 0.84 | 0.942 |
| | 0 | 27 (28.72%) | 41 (28.67%) | 1.000 | 18 (27.69%) | 7 (21.88%) | 0.712 | 39 (27.08%) | 66 (26.69%) | 0.958 |
| | 1 | 45 (47.87%) | 76 (53.15%) | 0.508 | 34 (52.31%) | 18 (56.25%) | 0.881 | 67 (46.53%) | 122 (48.61%) | 0.769 |
| | 2 | 18 (19.15%) | 21 (14.69%) | 0.467 | 10 (15.38%) | 3 (9.38%) | 0.617 | 28 (19.44%) | 47 (18.73%) | 0.966 |
| | 3 | 4 (4.26%) | 5 (3.50%) | 1.000 | 3 (4.62%) | 4 (12.50%) | 0.320 | 10 (6.94%) | 16 (6.37%) | 0.993 |
| MVD | | 0.95 ± 0.78 | 0.89 ± 0.75 | 0.566 | 0.94 ± 0.80 | 1.16 ± 0.87 | 0.246 | 1.05 ± 0.84 | 1.04 ± 0.83 | 0.922 |
| | 0 | 28 (29.79%) | 45 (31.47%) | 0.896 | 20 (30.77%) | 6 (18.75%) | 0.311 | 39 (27.08%) | 67 (26.69%) | 1.000 |
| | 1 | 46 (48.94%) | 73 (51.05%) | 0.853 | 32 (49.23%) | 19 (59.38%) | 0.469 | 68 (47.22%) | 123 (49.00%) | 0.813 |
| | 2 | 17 (18.09%) | 21 (14.69%) | 0.605 | 10 (15.38%) | 3 (9.38%) | 0.617 | 28 (19.44%) | 46 (18.33%) | 0.889 |
| | 3 | 3 (3.19%) | 4 (2.80%) | 1.000 | 3 (4.62%) | 4 (12.50%) | 0.320 | 9 (6.25%) | 15 (5.98%) | 1.000 |
| Volume loss | | 0.55 ± 0.68 | 0.48 ± 0.59 | 0.412 | 0.63 ± 0.67 | 0.66 ± 0.77 | 0.876 | 0.72 ± 0.71 | 0.57 ± 0.69 | **0.045** |
| | 0 | 51 (54.26%) | 81 (56.64%) | 0.819 | 31 (47.69%) | 16 (50.00%) | 1.000 | 61 (42.36%) | 135 (53.78%) | **0.037** |
| | 1 | 35 (37.23%) | 55 (38.46%) | 0.957 | 27 (41.54%) | 12 (37.50%) | 0.872 | 63 (43.75%) | 89 (35.46%) | 0.128 |
| | 2 | 7 (7.45%) | 7 (4.90%) | 0.594 | 7 (10.77%) | 3 (9.38%) | 1.000 | 19 (13.19%) | 26 (10.36%) | 0.491 |
| | 3 | 1 (1.06%) | 0 (0%) | 0.832 | 0 (0%) | 1 (3.13%) | 0.716 | 1 (0.69%) | 1 (0.40%) | 1.000 |
| White matter hyperintensity/lesion | | 0.62 ± 0.50 | 0.38 ± 0.28 | 0.338 | 1 ± 0.38 | 0.33 ± 0.24 | 0.645 | 0.41 ± 0.39 | 0.49 ± 0.43 | 0.333 |
| | 0 | 13 (13.83%) | 20 (13.99%) | 1.000 | 2 (3.08%) | 4 (12.50%) | 0.173 | 23 (15.97%) | 41 (16.33%) | 1.000 |
| | 1 | 4 (4.26%) | 12 (8.39%) | 0.329 | 2 (3.08%) | 2 (6.25%) | 0.845 | 6 (4.17%) | 21 (8.37%) | 0.166 |
| | 2 | 3 (3.19%) | 0 (0%) | 0.120 | 2 (3.08%) | 0 (0%) | 0.808 | 2 (1.39%) | 3 (1.20%) | 1.000 |
| | 3 | 1 (1.06%) | 0 (0%) | 0.832 | 0 (0%) | 0 (0%) | NA | 1 (0.69%) | 2 (0.80%) | 1.000 |

COVID-19, Coronavirus Disease 2019; CT, computed tomography; MRI, magnetic resonance imaging; MVD, microvascular disease; NA, not applicable; p-val, *p*-value; SD, standard deviation.

for both study cohorts due to advanced age and high prevalence of comorbidities, although it is higher than reported in some studies [33–36]. Readmission could be due to age-related illness or medical conditions exacerbated by COVID-19. The neurological cohort had a higher readmission rate than the control cohort, possibly suggesting the presence of a higher burden of disease during COVID-19 progression within this cohort.

The incidence of stroke was 2% to 6% and of heart attack was 4% to 5% in both groups. The incidence of MACE after discharge (16% and 20%) was higher than other have reported previously in COVID-19 who did not have neurological issues [24]. A few studies have suggested that COVID-19 exerts long-term cardiovascular effects [22,37–40], consistent with a disease

**Table 6. Changes in imaging findings between baseline (pre- or during COVID-19 hospitalization) and follow-up (post-COVID-19 hospitalization).** Pre-covid was anytime prior to each patient's individual COVID-19 infection, during COVID was during COVID-19 hospitalization, and post COVID-19 imaging was anytime after discharge and up to January 23, 2023. Only patients with both a baseline and follow-up scan were included. Imaging studies were scored as 0 (normal or no abnormality), 1 (mild abnormality), 2 (moderate abnormality), and 3 (severe abnormality). Mean ± SD or $n$ (%). $\chi^2$ used to compare categorical variables and two-tailed $t$ test use to compare continuous variables between groups.

| | Neurological (N = 112) | | Control (N = 157) | | |
|---|---|---|---|---|---|
| | | $p$-val (neurological) | | $p$-val (control) | $p$-val (neurological vs. control) |
| **Changes in incidence for** | | | | | |
| Hemorrhage | 3 (2.68%) | 0.245 | 3 (1.91%) | 0.246 | 0.999 |
| Active stroke | 7 (6.25%) | **0.021** | 6 (3.82%) | **0.039** | 0.531 |
| Prior stroke | 14 (12.50%) | **<0.001** | 15 (9.55%) | **<0.001** | 0.570 |
| Mass effect | 5 (4.46%) | 0.070 | 4 (2.55%) | 0.131 | 0.605 |
| Microhemorrhage | 8 (7.14%) | **0.012** | 10 (6.37%) | **0.004** | 0.998 |
| **Changes in scores for** | | | | | |
| White matter change | 0.12 ± 0.44 | **0.006** | 0.18 ± 046 | **<0.001** | 0.219 |
| MVD | 0.15 ± 0.41 | **<0.001** | 0.18 ± 046 | **<0.001** | 0.538 |
| Volume loss | 0.15 ± 0.41 | **<0.001** | 0.17 ± 0.42 | **<0.001** | 0.787 |
| White matter Hyperintensity/lesion | −0.04 ± 0.63 | 0.459 | 0.03 ± 0.42 | 0.452 | 0.310 |

COVID-19, Coronavirus Disease 2019; MVD, microvascular disease; $p$-val, $p$-value.

that affects the cardiovascular system and thus can result in MACE after severe infection warranting hospitalization. Approximately 4% to 5% of both cohorts experienced COVID-19 reinfection. This rate of reinfection is slightly higher than what is reported elsewhere [41,42]. The higher rate of reinfection may be attributed to the urban setting of congested environs, high rates of comorbidities, and healthcare disparities associated with lower socioeconomic status [43].

Of those who died post-discharge, more than half died within the first 0.5 years in both groups. The cumulative mortality rates of the neurological and control cohorts at 3 years post-discharge were 14% and 8%, respectively. The marked mortality rate differences between groups are highlighted by the Kaplan–Meier analysis. Those who died at follow-up in both groups were 8 years older and more likely to be of male sex as compared to survivors. Older patients may be more prone to exhibiting early neurologic symptoms because of differential degrees of accelerated pathological aging phenotypes that preferentially target brains that may harbor more limited degrees of cognitive resilience. Patients presenting with early neurologic compromise could be a harbinger of susceptibility for higher mortality across other disease states. Male sex has been previously reported to have worse acute in-hospital outcomes, including higher rate of multi-organ injury, critical care illness, and in-hospital mortality [44–49].

The univariate logistic regression model identified discharge disposition to be the top risk factor for post-discharge mortality, followed by CHF, COVID-19 severity score, belonging to the neurological cohort, and age. CHF was the only comorbidity that was significantly associated with post-discharge mortality. Patients in the control cohort who had more severe COVID-19 disease were also more likely to die post-discharge. Belonging to the neurological cohort was also an independent risk factor for post-discharge mortality. It is not surprising that advanced age is associated with higher post-discharge mortality, but advanced age ranked lower than other variables mentioned above. Note that OR for male sex was statistically insignificant, and we predicted that large sample sizes could result in significant findings. Taken together, these findings underscore the independent risk factors that contributed to post-

discharge mortality and notably belonging to the neurological cohort is a significant independent risk factor.

The causes of death were similar between neurological and control cohorts, consistent with findings using univariate logistic regression in which belonging to the neurological cohort was an important but not the most important associative variable of post-COVID-19 discharge mortality. The primary known causes of death in both the neurological and control cohorts were heart disease, sepsis, influenza and pneumonia, COVID-19, and ARDS. Sepsis, pneumonia, and ARDS might be related to or be triggered by COVID-19, although they could also be a result of other medical events. It is possible that COVID-19 as a cause of death was underestimated because of imprecise categorization in the death certificates. Note that about one-third of the causes of death were specified as unknown on the death certificates. To our knowledge, it is possible that some patients died of senescence and no primary cause of death was noted.

The age of the patient was taken into consideration when assessing neuroradiological findings. Imaging findings of patients in the control and neurological cohorts displayed differential profiles of abnormalities that were consistent with age and comorbidities in this population. The differences in radiological findings between baseline and follow-up showed age-related effects. However, there were generally no group differences in either qualitative or score-based findings, except that the neurological cohort showed greater volume loss post-COVID-19 compared to controls.

Several case reports and a few cohort studies have identified reduction in gray matter thickness, ischemic stroke, decrease in global brain size, cerebral microstructural changes, and persistent WM changes associated with COVID-19 [50–53]. There is likely some reporting bias in case or case-series studies as positive clinical imaging findings associated with COVID-19 are more likely to be noted. Most of these studies do not compare findings to baseline [50–53], which makes it difficult to discern whether imaging abnormalities were preexisting or a consequence of COVID-19 disease. None of these studies employ a scoring system to accentuate the degree of abnormality. To the best of our knowledge, our study is novel because of its large and diverse patient population, long follow-up times, the use of a scoring system, and comparison between baseline and follow-up studies up to 3 years post-discharge. It is possible that COVID-19–related changes in brain anatomy and structure could take time to manifest, and we predict that some patients with COVID-19 will likely experience accelerated aging and higher incidence of age-related disorders. Brain imaging is important because it could provide neural correlates of post-COVID-19 neurological sequela, which include, but are not limited to, neurological symptoms, neurocognitive deficits, fatigue, memory loss, anxiety, depression, and post-traumatic stress disorder [8,54–56].

Taken together, these current observations contribute new insights concerning our understanding of the longitudinal effects of neurological involvement in long COVID. In terms of nervous system involvement, persistence, and evolution, there are likely bidirectional interactions between the nervous and the immune systems that orchestrate a composite pro-inflammatory, hypercoagulable, hypoxemic, and immune dysregulated state [7]. In addition, these dynamic processes could contribute to accelerated brain aging, stress pathway-mediated neural injury responses, features of traumatic encephalopathy, demyelination, neurodegeneration, and accompanying preferential cortical atrophy as we have identified in this study. Moreover, multiple communication routes between the central and peripheral nervous systems and the body in long COVID create systemic organ system, tissue and cellular interfaces that impair organismal homeostasis to give rise to persistent deregulated regenerative and plasticity responses [7,57]. These pathological processes promote chronic multi-organ dysfunction and can lead to a spectrum of stressor states that predispose to organ fibrosis, tissue degeneration and even dysplastic and neoplastic conditions with associated metabolic derangement,

immune dysregulation, inflammatory processes, and additional features of SARS-CoV-2/COVID-19-mediated dyshomeostasis syndrome, including proteotoxicity and protean epigenetic derangements [58–60]. Such biological contingencies suggest that the multifactorial nature of the neurological manifestations of COVID-19 may put patients at higher risk of long-term functional disabilities and death as we have currently observed. Importantly, our increasing understanding of the nature of the deregulation of dynamic nervous system-systemic crosstalk displayed in response to SARS-CoV-2 may allow us to devise innovative and interdisciplinary mitigation strategies to alleviate the long-term sequelae preferentially caused by early neurological involvement in COVID-19.

This is one of the largest cohort studies of imaging findings and one of the longest follow-up studies of COVID-19 survivors. This study however has several limitations. Our patient cohort was limited to patients infected with COVID-19 during the first wave of the pandemic, when hospitals were overburdened, COVID-19 vaccines were not yet available, and COVID-19 treatments were limited.

The patient profiles (i.e., age composition) might differ from those of subsequent waves. Additionally, the selection of patients was subjective and may have introduced selection bias. In building the control cohort, propensity scoring only factored in age and COVID-19 severity to remain consistent with the index study. Despite all patients being infected with COVID-19 prior to vaccine development, vaccination status post-discharge may have affected long-term patient outcomes; given the unreliable data reporting on vaccines in our EMR, vaccination status was excluded from the study.

Although the attrition is low (12%), patients who did not return to our health system could not be studied. While it is possible that returning patients were more likely to have more severe COVID-19, our patient data consisted of those who returned for any medical reasons, including regular checkups. On the other hand, those who did not return might have expired. Our current study was not powered to address differences due to race and ethnicity because both study cohorts consisted of large proportions of blacks and Hispanics but lower proportions of other races and ethnic groups. Imaging sample sizes were small because not all patients had imaging performed at all 3 time points. The mixture of MRI and CT may have different sensitivities for various accompanying pathologies and more sophisticated imaging techniques may be warranted. Future studies should compare results with those of the general population without SARS-CoV-2 infection. Other factors such as reinfection, COVID-19 vaccination status, and influenza vaccination status could affect long-term outcomes. Most patients with COVID-19 were early in the pandemic before vaccines became available. It is likely COVID-19 vaccines and improved COVID-19 treatments will likely reduce long-term neurological sequela. Another limitation is that many controls did not have imaging results and thus findings need to be interpreted with caution. As with any retrospective study, there could be other unintended patient selection biases and unaccounted for confounders. Respiratory parameters such as the need for mechanical ventilation, duration of mechanical ventilation, need for tracheostomy, major life-sustaining support including dialysis, extracorporeal membrane oxygenation are needed to better understand if the effect on discharge, readmission, stroke and mortality is being completely driven by neurological symptoms or other confounders [61–64]. However, during the early chaotic phase of COVID-19 pandemic, many of these life-saving treatments were not consistently applied across patients. With overburdened hospital conditions, many general hospital floors, hallways, and emergency rooms became makeshift ICU rooms, resulting in some inaccurate documentation. Extracorporeal membrane oxygenation and dialysis in acute COVID-19 were rare in our cohort.

Patients with significant neurological findings during COVID-19 hospitalization were more likely to have worse outcomes at 3-year follow-up compared to propensity matched

controls. Improved understanding of the long-term outcomes of patients with COVID-19 with neurological involvement could help to develop effective screening methods and innovative interventions to address the potentially high burden of care among these COVID-19 survivors.

## Supporting information

**S1 Checklist. STROBE Statement.**
(DOCX)

**S1 Table. Proof of propensity match between neurological and control cohorts.**
(DOCX)

**S2 Table. Laboratory values of patients in the neurological and control cohorts at COVID-19 hospitalization admission and most recent follow-up.**
(DOCX)

**S3 Table. Number of patients treated in the ICU or placed on IMV during hospitalization for each outcome.**
(DOCX)

**S1 Fig. Flow diagram of initial and final sample sizes.**
(TIFF)

**S2 Fig.** (A) Distribution of COVID-19 severity score in the neurological and control cohorts (survivors after COVID-19 hospitalization discharge). Percent of patients in the neurological and control cohorts who (B) were readmitted, (C) had stroke, (D) had heart attack, (E) had MACE, and (F) died after discharge from COVID-19 hospitalization.
(TIFF)

## Author Contributions

**Conceptualization:** Anna Eligulashvili, Jimmy S. Lee, Kevin Hsu, Imanyah Hilliard, Kristen Lee, Arleen Li, Judy Yee, David J. Altschul, Emad Eskandar, Mark F. Mehler, Tim Q. Duong.

**Data curation:** Moshe Gordon, Jimmy S. Lee, Jeylin Lee, Shiv Mehrotra-Varma, Jai Mehrotra-Varma, Kevin Hsu, Imanyah Hilliard, Kristen Lee, Arleen Li, Muhammed Amir Essibayi, David J. Altschul, Emad Eskandar, Mark F. Mehler, Tim Q. Duong.

**Formal analysis:** Anna Eligulashvili, Moshe Gordon, Jeylin Lee, Shiv Mehrotra-Varma, Jai Mehrotra-Varma.

**Investigation:** Anna Eligulashvili, Jimmy S. Lee, Kevin Hsu, Imanyah Hilliard, Kristen Lee, Arleen Li, Muhammed Amir Essibayi, Judy Yee, David J. Altschul, Emad Eskandar, Mark F. Mehler, Tim Q. Duong.

**Methodology:** Anna Eligulashvili, Jimmy S. Lee, Kevin Hsu, Imanyah Hilliard, Kristen Lee, Arleen Li, Muhammed Amir Essibayi, Judy Yee, David J. Altschul, Emad Eskandar, Mark F. Mehler, Tim Q. Duong.

**Resources:** Judy Yee, Tim Q. Duong.

**Software:** Moshe Gordon, Jeylin Lee, Shiv Mehrotra-Varma, Jai Mehrotra-Varma.

**Supervision:** Judy Yee, Tim Q. Duong.

**Validation:** Anna Eligulashvili, Moshe Gordon, Jimmy S. Lee, Jeylin Lee, Shiv Mehrotra-Varma, Jai Mehrotra-Varma, Kevin Hsu, Imanyah Hilliard, Kristen Lee, Arleen Li, Muhammed Amir Essibayi, Judy Yee, David J. Altschul, Emad Eskandar, Mark F. Mehler, Tim Q. Duong.

**Writing – original draft:** Anna Eligulashvili, Tim Q. Duong.

**Writing – review & editing:** Anna Eligulashvili, Moshe Gordon, Jimmy S. Lee, Jeylin Lee, Shiv Mehrotra-Varma, Jai Mehrotra-Varma, Kevin Hsu, Imanyah Hilliard, Kristen Lee, Arleen Li, Muhammed Amir Essibayi, Judy Yee, David J. Altschul, Emad Eskandar, Mark F. Mehler, Tim Q. Duong.

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
