## [Editor Report · Decision Letter 0]

23 Jun 2023

Dear Dr Duong, 

Thank you for submitting your manuscript entitled "Long-term outcomes of hospitalized SARS-CoV-2/COVID-19 patients with and without neurological involvement: 3-year follow-up assessment" for consideration by PLOS Medicine.

Your manuscript has now been evaluated by the PLOS Medicine editorial staff and I am writing to let you know that we would like to send your submission out for external peer review.

Please re-submit your manuscript within two working days, i.e. by Jun 27 2023 11:59PM.

Kind regards,

Alexandra Schaefer, PhD

Associate Editor

PLOS Medicine

---

## [Decision Letter · Decision Letter 1]

29 Aug 2023

Dear Dr. Duong,

Thank you very much for submitting your manuscript "Long-term outcomes of hospitalized SARS-CoV-2/COVID-19 patients with and without neurological involvement: 3-year follow-up assessment" (PMEDICINE-D-23-01747R1) for consideration at PLOS Medicine. 

Your paper was evaluated by an associate editor and discussed among all the editors here. It was also discussed with an academic editor with relevant expertise, and sent to independent reviewers, including a statistical reviewer. The reviews are appended at the bottom of this email and any accompanying reviewer attachments can be seen via the link below:

[LINK]

In light of these reviews, I am afraid that we will not be able to accept the manuscript for publication in the journal in its current form, but we would like to consider a revised version that addresses the reviewers' and editors' comments. Obviously we cannot make any decision about publication until we have seen the revised manuscript and your response, and we plan to seek re-review by one or more of the reviewers. 

We expect to receive your revised manuscript by Sep 19 2023 11:59PM. Please email us (plosmedicine@plos.org) if you have any questions or concerns.

We look forward to receiving your revised manuscript. 

Sincerely,

Alexandra Schaefer, PhD

PLOS Medicine

plosmedicine.org

GENERAL COMMENTS

Please respond to all editor and reviewer comments.

Please cite the reference numbers in square brackets (e.g., “We used the techniques developed by our colleagues [19] to analyze the data”). Citations should be preceding punctuation.

Please cite your Supporting Information as outlined here: https://journals.plos.org/plosmedicine/s/supporting-information

ACADEMIC EDITOR COMMENTS

The manuscript could use more information on whether the participants were lost to follow up had different baseline characteristics, and why those who died during the hospitalization were excluded. The latter is relevant to the part regarding imaging, and how imaging findings or the presence of neurological injury led to withdrawal of care versus not withdrawing care, and those who did not have withdrawal could have been different from those of other centers who had imaging. Lastly there should be some context to this, any reason for admission is going to have worse outcomes in the long term when there is neurological dysfunction so that the findings may not be that surprising and it will be difficult to know if there is something unique about COVID with neurological injury as opposed to any other primary reason for admission.

EDITORIAL COMMENTS

We feel that the manuscript could benefit from being more focused and from exploring and discussing the implications of the study results in more depth.

EDITOR-IN-CHIEF COMMENTS 

Please include reporting of acute and long-term neurological outcomes in your manuscript or discuss why this was not done in detail. In addition, please provide a detailed description of the propensity score matching and the optimisation of the matching used.

DATA AVAILABILITY STATEMENT

PLOS Medicine requires that the de-identified data underlying the specific results in a published article be made available, without restrictions on access, in a public repository or as Supporting Information at the time of article publication, provided it is legal and ethical to do so. Please see the policy at http://journals.plos.org/plosmedicine/s/data-availability and FAQs at http://journals.plos.org/plosmedicine/s/data-availability#loc-faqs-for-data-policy

The Data Availability Statement (DAS) requires revision. For each data source used in your study: 

For studies in which a novel model is central to the manuscript's findings, as is the case here, authors are responsible for providing the source code needed to replicate the study's findings in a repository (such as GitHub, SourceForge or Bitbucket) or a cloud computing service (such as Code Ocean). Protection of authors’ intellectual property will not be cause for exception. Please explain in the manuscript’s Data Availability Statement how readers can access the shared code.

ABSTRACT

Please structure your abstract using the PLOS Medicine headings (Background, Methods and Findings, Conclusions). The Methods and Findings sections should be combined into one section, “Methods and findings”.

Please ensure that all numbers presented in the abstract are present and identical to numbers presented in the main manuscript text.

PLOS Medicine requests that main results are quantified with 95% CIs as well as p values. When reporting p values please report as p<0.001 and where higher as the exact p value p=0.002, for example. Please provide specific P values in place of [NS or P>.05]. For the purposes of transparent data reporting, if not including the aforementioned please clearly state the reasons why not.

Please include any important dependent variables that are adjusted for in the analyses.

Throughout, suggest reporting statistical information as follows to improve clarity for the reader “22% (95% CI [13%,28%]; p</=)”. Please amend throughout the abstract and main manuscript.

Please note the use of commas to separate upper and lower bounds, as opposed to hyphens as these can be confused with reporting of negative values.

Please define all abbreviations used for statistical reporting at first use (HR, CI, OR).

When a p value is given, please specify the statistical test used to determine it.

Please include the study design and study population, i.e. please provide brief demographic details of the study population (e.g. sex, age, ethnicity, etc).

In the last sentence of the Abstract Methods and Findings section, please describe the main limitation(s) of the study's methodology.

l.71: Please define ‘vs’ at first use.

l.86: Please change ‘have’ to ‘had’.

AUTHOR SUMMARY

At this stage, we ask that you include a short, non-technical Author Summary of your research to make findings accessible to a wide audience that includes both scientists and non-scientists. The Author Summary should immediately follow the Abstract in your revised manuscript. This text is subject to editorial change and should be distinct from the scientific abstract. Please see our author guidelines for more information: https://journals.plos.org/plosmedicine/s/revising-your-manuscript#loc-author-summary.

The summary should include 2-3 single sentence, individual bullet points under each of the questions.

It may be helpful to review currently published articles for examples which can be found on our website here https://journals.plos.org/plosmedicine/

INTRODUCTION

Please address past research and explain the need for and potential importance of your study. Indicate whether your study is novel and how you determined that. If there has been a systematic review of the evidence related to your study (or you have conducted one), please refer to and reference that review and indicate whether it supports the need for your study. 

Please conclude the Introduction with a clear description of the study question or hypothesis.

METHODS AND RESULTS

Please ensure that the study is reported according to the STROBE guideline, and include the completed STROBE checklist as Supporting Information. Please add the following statement, or similar, to the Methods: “This study is reported as per the Strengthening the Reporting of Observational Studies in Epidemiology (STROBE) guideline (S1 Checklist).“

Did your study have a prospective protocol or analysis plan? Please state this (either way) early in the Methods section.

For all observational studies, in the manuscript text, please indicate: (1) the specific hypotheses you intended to test, (2) the analytical methods by which you planned to test them, (3) the analyses you actually performed, and (4) when reported analyses differ from those that were planned, transparent explanations for differences that affect the reliability of the study's results. If a reported analysis was performed based on an interesting but unanticipated pattern in the data, please be clear that the analysis was data-driven.

PLOS Medicine requests that main results are quantified with 95% CIs as well as p values. When reporting p values please report as p<0.001 and where higher as the exact p value p=0.002, for example. For the purposes of transparent data reporting, if not including the aforementioned please clearly state the reasons why not.

Please include any important dependent variables that are adjusted for in the analyses.

Suggest reporting statistical information as detailed above – see under ABSTRACT

Please define "lost to follow-up" as used in this study. Other reasons for exclusion should be defined.

Please define the length of follow up (eg, in mean, SD, and range).

Please provide the actual numbers of events for the outcomes, not just summary statistics or ORs.

Please present numerators and denominators for percentages, at least in the Tables [not 

necessarily each time they're mentioned].

l.115: Please define ‘PCR’.

l.161: Please insert a comma following “lymphocyte count (Lymph)”.

ll.166-167: When describing age, please ensure to add a unit, such as ‘years’. Please revise throughout the entire manuscript.

l.168: Please add a unit for mean arterial pressure (MAP).

l.169: Please add a unit for blood urea nitrogen (BUN).

l.170: Please add a unit for C-reactive protein (CRP).

ll.181-182 please change to: “Major findings on both CT and MRI, were documented for their absence or presence of hemorrhage, active stroke, prior stroke, mass effect, and microhemorrhage.”

l.186: Please clarify that you refer to COVID-19 hospitalization. Please revise throughout the entire manuscript.

l.195: The statement “hospital readmission post discharge January 23, 2023 (3 years post COVID-19).” might be misleading and we suggest changing it to “hospital readmission post discharge until January 23, 2023 (3 years post COVID-19).”.

l.202: Please change “confidential interval” to “confidence interval”.

Please provide the name(s) of the institutional review board(s) that provided ethical approval.

ll.222-223: Please remove the “Data Availability Statement” from the main manuscript. This information should only be included in the corresponding section in the online submission form.

l.227/228: Please change ‘our’ to ‘the’.

l.239: Please add a unit for age.

l.240: Please exchange ‘gender’ with ‘sex’.

ll.252-254 suggest: “Mortality rates post COVID-19 hospitalization were also higher in the neurological cohort compared to the control cohort at 0.5 years (7.73% vs 3.67%, p=0.0004), 1 year (9.18% vs 4.92%, p=0.0009), and 3 years (14.01% vs 7.84%, p=0.0001) follow-up.” or similar.

l.262: Please change the reference to ‘Supplementary Figure 1B-F’.

l.278: Please change ‘compared’ to ‘compares’.

l.279: Please change ‘are’ to ‘were’.

ll.278-284: When possible, please report the quantified main results with 95% CIs as well as p values. Please revise throughout the entire manuscript.

ll.288-291 suggest: “These variables included belonging the neurological cohort (OR=1.802 (1.237, 2.608), p=0.002), discharge disposition (OR=1.508, 95% CI=(1.276, 1.775), p<0.0001), congestive heart failure (OR=2.281 (1.429, 3.593), p=0.0004), COVID-19 severity score (OR=1.177 (1.062, 1.304), p=0.002), and age (OR=1.027 (1.010, 1.044), p=0.002).”

ll.291-292: The term "trend" is used to refer to a nonsignificant P value. The term trend should be used only when the test for trend has been conducted. Please revise accordingly.

l.305: Please add ‘control’ to “[…] score of 0: 42.36% neurological vs. 53.78%, p=0.037).”.

ll.295-314: Please ensure that you are clear when describing the groups you are referring to, as you may vary the comparison throughout the manuscript, e.g. 'survivors versus non-survivors' and 'neurological versus control cohort'.

ll.308-314: Please be sure to include details of the statistical results (and not only for the first bracket), i.e. the groups to which the results presented belong (neurological, control, etc.).

DISCUSSION

Please present and organize the Discussion as follows: a short, clear summary of the article's findings; what the study adds to existing research and where and why the results may differ from previous research; strengths and limitations of the study; implications and next steps for research, clinical practice, and/or public policy; one-paragraph conclusion.

Please remove any subheadings.

l.337: Please change ‘our’ to ‘the’.

l.348: In line 338 you refer to the whole group of participants (neurological and control cohorts) as one cohort (our cohort), whereas here you refer to the whole group as 'both cohorts'. Please revise to ensure clarity (throughout the entire manuscript).

l.359: Please add a reference for “This is widely seen in other medical conditions such as urosepsis.”.

l.361/363: Please exchange ‘gender’ with ‘sex’. Please revise throughout the entire manuscript.

ll.367-368: Please change to: “Belonging to the neurological cohort was also an independent risk factor for post-discharge mortality.”

l.372: Please ensure that you use a consistent spelling when writing 'post discharge' ('post discharge' or 'post-discharge'). Please revise throughout the entire manuscript.

ll.388-390: Please change to “However, there were generally no group differences in either qualitative or score-based findings, except that the neurological cohort showed greater volume loss post-COVID-19 compared to controls.”.

l.397: Please temper claims of primacy of results by stating, "to our knowledge" or something similar.

ll.413-420: Please add according references.

ll.442-444 suggest: “The mixture of MRI and CT may have different sensitivities for various accompanying pathologies and more sophisticated imaging techniques may be warranted.” 

Please remove the ‘Study funding’ section. This information should only be included in the corresponding section in the online submission form.

TABLES

Please provide titles and descriptions for all tables (including those in Supporting Information files).

Please define abbreviations used in the table of each table (including those in Supporting Information files).

When reporting p-values in tables (including those in Supporting Information files), please include a separate column (in each table) with the p-values and report as <0.001 or the exact p-value where higher and remove all reference to asterisks and p-value thresholds.

Table 1: Please define ‘SD’, ‘COPD’, ‘CHF’, CKD’, ‘SNF’, ‘MACE’.

Table 1: For consistency, please change the column header to ‘Neurological Cohort’.

Table 2: Please define ‘ARDS’, ‘GI’, ‘ICH’

Table 3: Please change ‘year old’ to ‘years old’. 

Table 3: Please define ‘SD’, ‘COPD’, ‘SNF’, ‘OR’, ‘CI’, ‘p-val’.

Table 3: In (B), please add ‘Sex’ to ‘Male’.

Table 4: Please note that the main text refers to 'Table 4A' and 'Table 4B', whereas the actual table is divided into 'Table 4' and 'Table 4B'.

Table 4: In the title you refer to ‘COVID-19 infection’, whereas in the manuscript you refer to ‘COVID-19 hospitalization’. Please revise accordingly.

Table 4: For clarity, please describe ‘Studies’ as ‘Imaging studies’ or similar.

Table 4: Please define ‘MRI’, ‘CT’, ‘p-val’. We suggesting writing ‘p-value’ in full in the column header.

Table 4: Please define the meaning of the numbers following ‘±’ (e.g. standard deviation).

Table 4B: For clarity, please describe ‘Studies’ as ‘Imaging studies’ or similar.

Table 4B: In the table description or the column headers, please ensure to mention that you are referring to the different cohorts.

Table 4B: Please define the meaning of the numbers following ‘±’ (e.g. standard deviation).

Supplemental Table 1: Please ensure to define all abbreviations (including units) used in the table.

Supplemental Table 1: For consistency, please change the column header to ‘Neurological Cohort’.

FIGURES

For all Figures, please ensure that you have complied with our figures requirements http://journals.plos.org/plosmedicine/s/figures.

Please provide titles and legends for all figures (including those in Supporting Information files).

Please define abbreviations used in the table/figure legend of each figure and/or table (including those in Supporting Information files).

Please consider avoiding the use of red and green in order to make your figure more accessible to those with colour blindness.

Figure 2: In the Kaplan-Meier curve, please provide the number at risk for each time interval.

Figure 2: Please indicate in the figure caption the meaning of the whiskers.

Figure 2: Please show the axis beginning at zero. If this is not possible, please show a break in the axis.

Supplemental Figure 1: Please increase the size of the graphs.

Supplemental Figure 1: We suggest changing the axis title to ‘Patient Count (N)’.

REFERENCES

Please ensure that journal name abbreviations match those found in the National Center for Biotechnology Information (NCBI) databases (http://www.ncbi.nlm.nih.gov/nlmcatalog/journals) and are appropriately formatted and capitalised.

Please use the "Vancouver" style for reference formatting, and see our website for other reference guidelines https://journals.plos.org/plosmedicine/s/submission-guidelines#loc-references

Where website addresses are cited, please specify the date of access. 

Comments from the reviewers:

Reviewer #1: The present study investigated the long-term consequences of patients with and without significant neurological manifestations during the initial hospitalization for COVID-19 over a period of three years. Acute neurological manifestations are a common complication of acute COVID-19 disease. The results indicate that COVID-19 patients with neurological manifestations have worse long-term outcomes compared to matched control groups. These findings suggest the importance of close monitoring and timely interventions for COVID-19 patients with neurological manifestations. Raising awareness and implementing appropriate measures are crucial to minimize the long-term effects.

However, some important questions are not addressed or discussed in the study. Specifically, information about additional vaccinations over a three-year period is crucial and could have a significant impact on the longterm results. It would be worthwhile to at least discuss and consider how such vaccinations might affect the disease trajectory.

The gender-specific differences in the results are indeed very interesting and should be further investigated. A detailed analysis of the current literature on this topic could provide valuable insights and contribute to a better understanding of these differences.

Furthermore, it is important to mention that the observation period was limited to the early phase of the pandemic, focusing on the Wuhan variant, which was considered more aggressive compared to later virus variants. This fact should be taken into account in the discussion and the potential implications of this specific virus variant on the results of the manuscript should be discussed.

Overall, this is an interesting study with a large number of patients and a long observation period. However, the aforementioned points are important aspects that should be considered in a comprehensive discussion to further improve the understanding of the results. In the abstract, it should be mentioned which neurological disorders were investigated during the acute phase.

In Table 2, it should be clarified within which timeframe the causes of death after discharge were recorded. Did they encompass the entire three-year observation period or were they deaths that occurred immediately after discharge?

In line 240 on page 10, the gender composition should be specified more precisely to avoid misunderstandings.

Reviewer #2: Statistical review

This paper reports a retrospective cohort study assessing difference in post-discharge outcomes amongst individuals hospitalised for covid-19 with and without neurological symptoms. I had some comments on the statistical methods and reporting in the paper, which are listed below.

1. Abstract/results: For comparative results it would be useful to provide a between group difference measure (e.g odds ratio) and 95% CI as well as p-value.

2. Abstract methods: can a bit more be provided about the methods used to make inference? 'predictive models' is a big vague.

3. Abstract line 77 - could it be made clear that the percentages of mortality due to the listed reasons are across the two cohorts e.g. 'Across both cohorts, the major causes of death…'

4. Abstract line 79: were these results from univariable models or multivariable models? 

5. Abstract line 84: if a p-value is to be reported in the abstract, can the brain volume loss be reported similarly to other variables (i.e. difference in groups and more exact p-value).

6. Page 6: could it be clarified that the controls were hospitalised with covid over the same time period as the neurological cohort? If not, this would be somewhat of a limitation of the comparison.

7. Page 9: when were the listed secondary outcomes measured?

8. Page 9, line 199: I wasn't sure why this paragraph was included here rather than in 'Statistical analysis'? 

9. Page 9: It wasn't clear to me which variables were adjusted for in the logistic and Cox models.

10. Page 10 line 226: I wasn't sure how to interpret the numbers here. Does it indicate that the number 'returned to our health system' was the number in subsequent analyses, with 10.6% attrition representing the proportion discharged alive who weren't included in the analysis? It might be useful to add to the statistical analysis section which individuals were included in the analysis and how attrition was dealt with for the different types of outcomes.

11. Page 10 line 239: I don't think adding in 'p>0.05' for the results here is necessary. Perhaps including the p-values in Table 1 would be better?

12. Table 2 - I would recommend reporting estimated differences between arms, 95% CIs and p-values in this table.

13. Line 255 "cohort had a significantly lower survival probability than the control cohort at all timepoints" - I would change this to 'cohort had a significantly lower time to death than the control cohort'

14. Table 3B - are these results from a multivariable logistic regression? If so I would urge the authors to bear in mind the table 2 fallacy which would suggest that one should not overinterpret the individual variable odds ratios. If the main interest is in the neurological vs control comparison on mortality, then the model should be careful not to adjust for mediators (perhaps discharge disposition and congestive heart failure, unless that was from prior to hospitalisation?) otherwise the results are prone to bias from the table 2 fallacy. If the results are similar with not adjusting for these then this would be good to note. I would also recommend that the abstract and discussion addresses this issue (in particular the 'discharge to skilled nursing facilities').

15. Page 13: For the imaging analysis, it seems a high proportion of controls did not have results. As a result, these results seem less reliable to me - I would recommend caveating these results or mentioning assumptions made in the analysis. 

James Wason

Reviewer #3: I appreciate the opportunity to review the manuscript titled "Long-term outcomes of hospitalized SARS-CoV-2/COVID-19 patients with and without neurological involvement: 3-year follow-up assessment" by Eligulashvili. et. al. In this matched cohort study, the authors assessed long-term outcomes of patients admitted with COVID-19 based on presence or absence of neurological symptoms. The authors found patient with neurological symptoms where less likely to be discharge home, readmitted more, had higher incidence of mortality and stroke. 

I commend the authors for performing an extensive analysis of the dataset which significantly add to our current understanding of the deleterious effects of development of neurological symptoms in COVID-19, however they remain some major issues that needs to be resolved. 

Major concerns

1. Propensity score matching was based on age and covid-19 severity score.

The reference is for the previous article published by the group. Can you please reference the validation of this severity scoring system.

Also, it will be beneficial to know how well the propensity score matched the patient population. 

Another table in the supplement with information on the matched group will be highly helpful

2. A single bucket of neurological symptoms which range from stroke, seizure to impaired arousal and neuro-COVID-19 complex may limit which of these neurological symptoms are truly associated with worse outcome. A subgroup analysis based on structural vs non-structural pathology may be helpful. As it is difficult to ascertain that having delirium or vertigo during the stay is associated same long-term as having seizures, stroke or brain tumor. 

3. Flow diagram on the initial and final sample size will be very helpful. 

4. One of the most common causes of admission in COVID-19 was respiratory failure, however the data on their respiratory status was lacking. Providing and controlling for respiratory factor particularly need for mechanical ventilation, duration of mechanical ventilation, need for tracheostomy, major life sustaining support including dialysis, ECMO are needed to truly needed to better understand if the effect on discharge, readmission, stroke and mortality being seen is completely driving by neurological symptoms or other confounders.

5. The article title suggests long-term effect, however looking at the Kaplan-Meier survival analysis, a significant number of death are happening within 100 days. Most studies have used after 90 days or 6 months from discharge to determine long-term mortality (PMID: 21247589, PMID: 33643222). Please change the title and also, if the authors can provide more based on time period of 90 days post discharge to 1 yr, 2 yr and 3 yr, the information will be very helpful to the readers.

[LINK]

---

## [Decision Letter · Decision Letter 2]

30 Nov 2023

Dear Dr. Duong,

Thank you very much for submitting your manuscript "Long-term outcomes of hospitalized SARS-CoV-2/COVID-19 patients with and without neurological involvement: 3-year follow-up assessment" (PMEDICINE-D-23-01747R2) for consideration at PLOS Medicine. 

Thank you for your responses to the editors' and reviewers' comments. I have discussed the paper with my colleagues and the academic editor, and it has also been seen again by two of the original reviewers. While we appreciate your efforts to address the editors' and reviewers' comments, we feel that some of the comments have not been sufficiently addressed and require a more detailed response. In addition, we strongly feel that respiratory variables should be reported and included in your analysis. When submitting your revised paper, please include a detailed point-by-point response to the editorial comments.

The reviews and editorial comments are appended at the bottom of this email and any accompanying reviewer attachments can be seen via the link below:

[LINK]

In light of our remaining concerns, I am afraid that we will not be able to accept the manuscript for publication in the journal in its current form, but we would like to consider a revised version that addresses the reviewers' and editors' comments. Obviously we cannot make any decision about publication until we have seen the revised manuscript and your response, and we plan to seek re-review by one or more of the reviewers. 

We expect to receive your revised manuscript by Dec 21 2023 11:59PM. Please email me (aschaefer@plos.org) if you have any questions or concerns.

We look forward to receiving your revised manuscript. 

Sincerely,

Alexandra Schaefer, PhD

PLOS Medicine

plosmedicine.org

MAIN POINTS

1) The Editors agree with Reviewer 3 with regard to the reporting of respiratory variables. The initial review stated “One of the most common causes of admission in COVID-19 was respiratory failure, however the data on their respiratory status was lacking. Providing and controlling for respiratory factor particularly need for mechanical ventilation, duration of mechanical ventilation, need for tracheostomy, major life sustaining support including dialysis, ECMO are needed to truly needed to better understand if the effect on discharge, readmission, stroke and mortality being seen is completely driving by neurological symptoms or other confounders.” You responded that you do not have statistical power to attribute outcomes to different respiratory factors. However, because of the clinical importance of these variables, the Editors feel strongly that these variables should be included in the univariable analysis along with the other variables considered (noting that there were small numbers with regard to a number of other comorbidities, which would also limit the statistical power). 

2) Abstract – the Editors feel strongly that the data presented in the paper do not justify the claim to have identified a new syndrome that is distinct from Long COVID. Please remove the following text from the Abstract “…suggest the existence of a novel syndrome of persistent SARS-CoV-2/COVID-19 critical care illness associated with enhanced morbidity and mortality associated with initial neurological involvement that is distinct from Long COVID syndrome. These observations reinforce the notion of dynamic nervous system-systemic crosstalk in SARS-CoV-2/COVID-19 that give rise to a unique “dyshomeostasis syndrome”, with profound implications for a more nuanced perspective on disease pathogenesis and innovative therapeutic initiatives.” 

3) Introduction, lines 144-146. Please remove the statement, “Patients with neurological complications have been shown to have worse acute COVID-19 outcomes including a higher incidence of critical care illness and death compared to propensity-matched controls, thereby defining a novel COVID-19 syndrome distinct from ‘long COVID’”. As stated previously, the Editors do not feel that the published data merit the claim of identifying a new syndrome that is distinct from Long COVID. 

4) Abstract and Results: It is unclear why some of the 95% CIs are expressed as percentages. Eg. Lines 332-335: “Mortality rates post COVID-19 hospitalization were also higher in the neurological cohort compared to the control cohort at 0.5 years (OR=2.106, 95% CI=(1.28%, 6.84%); p<0.001), 1 year (OR=1.866, 95% CI=(1.22%, 7.30%), p<0.001), and 3 years (OR=1.787, 95% CI=(2.50%, 9.84%), p<0.001).” Please express these in the conventional way (e.g. “OR:1.222 (95% CI [1.101,1.789; p<0.001”).

5) Discussion, line 432-433: The Editors do not feel that the data support the conclusion regarding cause of death, and we ask that this sentence by modified/toned down or omitted: “Of those who died post-discharge, more than half died within the first 0.5 years in both groups, suggesting that most of these deaths were likely COVID-19 related.” Was COVID-19 listed as the primary cause of death for most/all of these patients?

SPECIFIC POINTS

DATA AVAILABILITY STATEMENT

Please update the manuscript online submission form with the details provided about data availability (ll.295-297) in the main manuscript and remove the statement from the main text.

ABSTRACT

1) Please define all numerical values (i.e. standard deviation (SD), standard error (SE) etc.). E.g., l.71 “69.71±15.80 years old”.

2) It is unusual to include ORs and p-values for differences between groups in baseline variables (e.g., sex, race/ethnicity). Please simply report these data as N (%). 

AUTHOR SUMMARY

1) Please define ‘MRI’ and ‘CT’.

2) Please remove the HR from the second bullet point in the “What did the researchers do and find” section. 

INTRODUCTION

ll.139-141: Please add references to support the following statement, “Additional studies suggest that a diffuse microvasculopathy may ensue with endothelial compromise, micro-infarctions, subsequent microhemorrhages, and microglial conglomerates with innate immune activation.”

METHODS

1) l.191: Please define ‘EMR’ at first use.

2) l.204: Please define all abbreviations at first use (e.g., ICD).

3) Please define all numerical values (i.e. standard deviation (SD), standard error (SE) etc.). E.g., l.306 “672.07±269.76 day” or l.307 “(0.138, 1037.413 days)”. Please revise throughout the entire manuscript.

4) ll.346-347: The term "trend" is used to refer to a nonsignificant P value. The term trend should be used only when the test for trend has been conducted. Please revise accordingly.

5) ll.352-353: “There were no group differences in cause of death between patients belonging to neurological and control cohorts (all p>0.05).“ – Since we require reporting of exact p-values, we suggest either reporting the exact p-value of each result or, since the results are all not statistically significant, removing "(all p>0.05)" and referring to the tabulated results. In all cases where "(all p>0.05)" is reported, please modify accordingly.

6) ll.358-365: If possible/applicable, please report results quantified with 95% CIs as well as p values. At minimum when stating significance, please report p-values. 

7) ll.372-373: The expression “approaching significance” seems to refer to a nonsignificant P value. As stated previously, please refrain from describing trends (or similar) when the test for trend has not been conducted. Please revise accordingly throughout the entire manuscript.

DISCUSSION

1) l.403: Please remove the HR.

2) l.424: What does "(16, 20%)“ mean?

3) l.437: Please exchange the term ‚elderly’ with ‘older’.

4) Please replace “COVID-19 patients” with “patients with COVID-19” (e.g. l.399). Please revise throughout the entire manuscript.

5) ll.450-451: Please refrain from describing trends (or similar) when the test for trend has not been conducted.

6) l.476: Please temper assertions of primacy ("Our study is novel because […]”) by adding ‘to the best of our knowledge’ or similar.

TABLES AND FIGURES

1) Table 1: It is unusual to include CI and p-values for differences between groups in baseline variables (e.g., sex, race/ethnicity). Please simply report these data as N (%). 

2) Table 3: Please define ‘yo’.

3) Supplemental Table: Please report the exact p-values.

4) Supplemental Figure 2 (A): Please ensure that the y-axis is identical for (A) to facilitate comparison.

Comments from the reviewers:

Reviewer #2: Thank you to the authors for addressing my previous comments well. I have no further issues to raise.

Reviewer #3: Thank you so much for providing me with opportunity to review the revised the manuscript.

I would have preferred to have more information available regarding the hospital course of this patients particularly need for respiratory support, given the retrospective nature, the author mentioned they were unable to obtain it and control for it.

I would prefer the author to include the following in limitation section.

The information on their hospital course particularly need for mechanical ventilation, tracheostomy and other life support were not available. Use the following literature to expand on the limitation (PMID: 33329361, PMID: 35994441) Our group also suggested that brain injury is common in ARDS (non-covid and COVID) during their stay  (PMID: 36094464, PMID: 37667080). In absence of data on severity of respiratory failure, MV, it is challenging to say that the poor outcome we see is all from neurological symptoms and not from interventions that caused both the neurological symptoms and poor outcome.

[LINK]

---

## [Editor Report · Decision Letter 3]

5 Jan 2024

Dear Dr. Duong,

Thank you very much for submitting your manuscript "Long-term outcomes of hospitalized patients with SARS-CoV-2/COVID-19 with and without neurological involvement: 3-year follow-up assessment" (PMEDICINE-D-23-01747R3) for consideration at PLOS Medicine. 

Thank you for letting us know that you are working on a careful review/extraction of the data we requested. The decision to revise has been made to allow you to include the respiratory data as discussed via email. We appreciate your efforts to comply with our requests. When you submit your revised paper, please include a detailed point-by-point response to the editors' comments.

Given the remaining requirements, I am afraid that we will not be able to accept the manuscript for publication in the journal in its current form, but we would be happy to consider a revised version that addresses the editors' comments. Of course, we cannot make a decision about publication until we have seen the revised manuscript and your response, and we plan to ask one or more of the reviewers to re-review it. 

We expect to receive your revised manuscript by Jan 26 2024 11:59PM. Please email me (aschaefer@plos.org) if you have any questions or concerns.

We look forward to receiving your revised manuscript. 

Sincerely,

Alexandra Schaefer, PhD

PLOS Medicine

plosmedicine.org

Please include the respiratory data as discussed by email.

Comments from the reviewers:

[LINK]

---

## [Decision Letter · Decision Letter 4]

16 Feb 2024

Dear Dr. Duong,

Thank you very much for re-submitting your manuscript "Long-term outcomes of hospitalized patients with SARS-CoV-2/COVID-19 with and without neurological involvement: 3-year follow-up assessment" (PMEDICINE-D-23-01747R4) for review by PLOS Medicine.

Thank you for your detailed response to the editors' and reviewers' comments. I have discussed the paper with my colleagues and the academic editor, and it has also been seen again by the statistical reviewer. The changes made to the paper were satisfactory to the reviewers. As such, we intend to accept the paper for publication, pending your attention to the editorial comments below in a further revision. When submitting your revised paper, please once again include a detailed point-by-point response to the editorial comments.

[LINK]

In revising the manuscript for further consideration here, please ensure you address the specific points made by each reviewer and the editors. In your rebuttal letter you should indicate your response to the reviewers' and editors' comments and the changes you have made in the manuscript. Please submit a clean version of the paper as the main article file. A version with changes marked must also be uploaded as a marked up manuscript file. Please also check the guidelines for revised papers at http://journals.plos.org/plosmedicine/s/revising-your-manuscript for any that apply to your paper. 

We ask that you submit your revision within 1 week (Feb 23 2024). However, if this deadline is not feasible, please contact me by email, and we can discuss a suitable alternative.

Please do not hesitate to contact me directly with any questions (aschaefer@plos.org). If you reply directly to this message, please be sure to 'Reply All' so your message comes directly to my inbox.

We look forward to receiving the revised manuscript.

Sincerely,

Alexandra Schaefer, PhD

Associate Editor 

PLOS Medicine

plosmedicine.org

Requests from Editors:

ABSTRACT

1) l.75: Please remove the age comparison as this variable was included in the propensity matching.

2) Please indicate in the Abstract what variables were used in the propensity score matching.

3) Please include ICU and IMV status in the Abstract (as done in line 310).

4) l.87: Please define ‘CI’ at first use.

5) l.91: Please define ‘OR’ at first use.

AUTHOR SUMMARY

The last bullet point under ‘What Do These Findings Mean?’ point should describe the main limitation of the study's methodology. 

METHODS AND RESULTS

1) ll.308-311: Please remove the age comparison as this variable was included in the propensity matching.

2) ll.315-317: “Some laboratory data at admission were worse than those at follow-up and laboratory data of the neurological cohort was worse than those of the control cohort.” – We feel this statement is rather vague. Please be more thorough and careful in the presentation of this data. It would be helpful to quantify these statements, e.g., how many of the measured laboratory values were worse at admission than at follow-up, and how many values were worse in the neurological cohort than in the control cohort (and vice versa).

3) ll.326-328: Please briefly describe the results of the number of patients treated in the ICU or with IMV (including numerical values). For example: “A higher proportion of patients in the neurological group who were readmitted to hospital or experienced MACE post-discharge were treated in the ICU or with IMV during their primary hospitalization compared with the control group (16 [6%] of 272 vs 14 [2%] of 728 for readmission; 5 [6%] of 85 vs 3 [2%] of 198 for MACE; table S3]” (or similar). Please note that we are not sure if we have interpreted the data correctly here (We would like you to present the following data: Of XXX who were readmitted post-COVID, xx (%) had been in the ICU and xx (%) had had IMV).

4) ll.351-354: “Both within the neurological and control cohorts, non-survivors were significantly older (neuro: 76.40±11.71 vs 67.85±15.85, p<0.001 and control: 76.56±11.23 vs 69.20±14.95, p<0.001) and had higher COVID-19 severity score (control: 4.74±2.05 vs 3.36±2.06, p<0.001) compared to survivors.” – Please revise the sentence as it currently reads that within the neurological cohort, non-survivors also had a higher COVID-19 severity score compared to survivors, which was not the case (p=0.088). We recommend breaking this into two sentences to avoid confusion.

5) ll.370-371: Please change to: "The number of patients who underwent imaging varied between pre-, intra-, and post-COVID-19."

DISCUSSION

1) ll.400-401: “the primary causes of death after discharge for both cohorts were heart failure, sepsis, influenza and pneumonia, COVID-19 and ARDS” – The primary cause of death after discharge was “unknown” – please revise accordingly.

2) ll.401-403: “patients who died post-discharge were significantly older, more likely to be of male sex, had higher COVID-19 severity score, and those were sent to skilled nursing facilities at discharge compared to survivors” – The results for sex were not significant. We feel that describing patients who died after discharge as more likely to be male may be misleading. Similarly, only for the control cohort was it true that patients who died post-charge had a higher COVID-19 severity score, which should be clarified here.

3) ll.402-403, please change to: “…and were more likely to have been discharged to skilled nursing facilities at discharge…”

4) ll.430-431: “...older and more likely to be of male sex as compared to survivors.” – Please clarify that this was not a significant finding.

5) ll.436-437: “Here we reported male sex also had worse long-term outcomes post COVID-19 discharge.” – This statement is not supported by the results, therefore please remove or re-phrase accordingly.

6) l.441: “Patients who had more severe COVID-19 disease were also more likely to die post-discharge.” – Please clarify that this only applies to the control cohort.

7) ll.486-493 “Moreover, multiple communication routes between the central...including proteotoxicity and protean epigenetic derangements.” – please provide references.

TABLES AND FIGURES

1) Table 4A: Please define ‘NA’. Please include dates (or time frames) for Pre, During, and Post COVID-19.

2) Table 4B: Please include dates (or time frames) for Pre, During, and Post COVID-19.

3) Figure 1: In the figure description, please define ‘Other’.

4) Supplemental Table 2: Please remove the definition of the p-values from the figure description as you have included the exact values.

5) Please ensure to include Supplemental Table 3 in the Supporting Information file.

REFERENCES

We noted that some of the references are not correctly formatted (e.g. [37]). Please thoroughly revise all references and ensure that they display first six authors, et al.

SOCIAL MEDIA

To help us extend the reach of your research, please provide any X (formerly known as Twitter) handle(s) that would be appropriate to tag, including your own, your coauthors’, your institution, funder, or lab. Please respond to this email with any handles you wish to be included when we tweet this paper.

Comments from Reviewers:

Reviewer #2: I've looked over the new results added and the changes to the paper look good. However I couldn't see Supplemental Table 3 in the provided document - perhaps the previous one was included?

One minor comment: Line 359: add 'no significant differences'.

[LINK]

General Editorial Requests

---

## [Editor Report · Decision Letter 5]

28 Feb 2024

Dear Dr Duong, 

On behalf of my colleagues and the Academic Editor, Joshua Z Willey, I am pleased to inform you that we have agreed to publish your manuscript "Long-term outcomes of hospitalized patients with SARS-CoV-2/COVID-19 with and without neurological involvement: 3-year follow-up assessment" (PMEDICINE-D-23-01747R5) in PLOS Medicine.

I appreciate your thorough responses to the reviewers' and editors' comments throughout the editorial process. We look forward to publishing your manuscript, and editorially there are only three remaining minor stylistic/presentation points that should be addressed prior to publication. We will carefully check whether the changes have been made. If you have any questions or concerns regarding these final requests, please feel free to contact me at aschaefer@plos.org.

Please see below the minor points that we request you respond to:

1) Please once again carefully revise all references for correct formatting. For example, references [38] or [41], still display more than six authors (PLOS uses first six authors, et al.).

2) The figure description of Supplemental Table 2 still contains the definitions of the p-values. Please remove “*p<0.05, **p<0.01, ***p<0.001 between the neurological and control cohorts. ^p<0.05, ^^p<0.01, ^^^p<0.001 between admission and follow-up lab values within the same cohort.”. 

3) We note that you have used the term "neurological patients/control patients" throughout the manuscript. We prefer the use of patient-centered language, i.e., terms such as "patients in the neurological/control cohort" or "patients with neurological manifestations" (or similar). Please revise accordingly.

PRESS

Sincerely, 

Alexandra Schaefer, PhD 

Associate Editor 

PLOS Medicine